# Aligning Optimization Trajectories with Diffusion Models for Constrained Design Generation

**Giorgio Giannone**
Massachusetts Institute of Technology
Technical University of Denmark
`ggiorgio@mit.edu`

**Akash Srivastava**
MIT-IBM Watson AI Lab
`akashsri@mit.edu`

**Ole Winther**
Technical University of Denmark
University of Copenhagen
`olwi@dtu.dk`

**Faez Ahmed**
Massachusetts Institute of Technology
`faez@mit.edu`

## Abstract

Generative models have significantly influenced both vision and language domains, ushering in innovative multimodal applications. Although these achievements have motivated exploration in scientific and engineering fields, challenges emerge, particularly in constrained settings with limited data where precision is crucial. Traditional engineering optimization methods rooted in physics often surpass generative models in these contexts. To address these challenges, we introduce Diffusion Optimization Models (DOM) and Trajectory Alignment (TA), a learning framework that demonstrates the efficacy of aligning the sampling trajectory of diffusion models with the trajectory derived from physics-based iterative optimization methods. This alignment ensures that the sampling process remains grounded in the underlying physical principles. This alignment eliminates the need for costly preprocessing, external surrogate models, or extra labeled data, generating feasible and high-performance designs efficiently. We apply our framework to structural topology optimization, a fundamental problem in mechanical design, evaluating its performance on in- and out-of-distribution configurations. Our results demonstrate that TA outperforms state-of-the-art deep generative models on in-distribution configurations and halves the inference computational cost. When coupled with a few steps of optimization, it also improves manufacturability for out-of-distribution conditions. DOM's efficiency and performance improvements significantly expedite design processes and steer them toward optimal and manufacturable outcomes, highlighting the potential of generative models in data-driven design.

## 1  Introduction

The remarkable progress in large vision [27, 20, 43, 85] and language models [30, 21, 78] has ushered in unparalleled capabilities for processing unstructured data, leading to innovations in multimodal and semantic generation [79, 67, 18, 80]. This momentum in model development has influenced the rise of Deep Generative Models (DGMs) in the realms of science [51, 61] and engineering [84, 101], especially in constraint-bound problems like structural Topology Optimization (TO [95]), offering potential to make the design process faster.

Many engineering design applications predominantly rely on *iterative* optimization algorithms. These algorithms break down physical and chemical phenomena into discrete components and incrementally enhance design performance while ensuring all constraint requirements are met. As an example, topology optimization aims to determine the optimal material distribution within a given design space,

37th Conference on Neural Information Processing Systems (NeurIPS 2023).

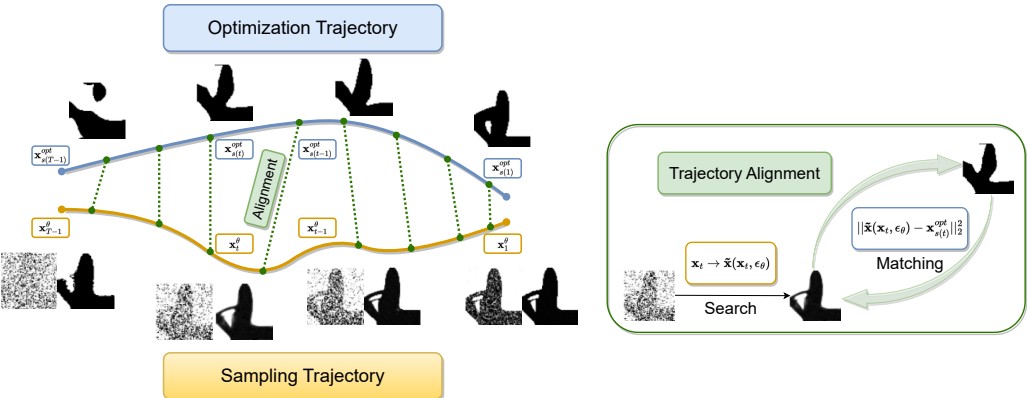

**(a)** Sampling and Optimization Trajectory       **(b)** Trajectory Alignment

**Figure 1:** Trajectory Alignment. Intermediate sampling steps in a Diffusion Optimization Model are matched with intermediate optimization steps. In doing so, the sampling path is biased toward the optimization path, guiding the data-driven path toward physical trajectories. This leads to significantly more precise samples.

under specified loads and boundary conditions, to achieve the best performance according to a set of defined criteria, such as minimum weight or maximum stiffness. Traditional iterative methods like SIMP [15] are invaluable but grapple with practical challenges, especially for large-scale problems, owing to their computational complexity.

Recent advancements have sought to address these challenges by venturing into learning-based methods for topology optimization, specifically DGMs. These models, trained on datasets comprising optimal solutions across varying constraints, can either speed up or even substitute traditional optimization processes. They also introduce a diverse set of structural topologies by tapping into large datasets of prior designs. The dual advantage of generating a variety of solutions and accommodating numerous design variables, constraints, and goals renders these learning-based methodologies especially enticing for engineering design scenarios.

However, purely data-driven approaches in generative design often fall short when benchmarked against optimization-based methods. While data-driven techniques might prioritize metrics such as reconstruction quality, these metrics might not adequately gauge adherence to engineering performance and constraint requirements. The core of this challenge lies in the data-driven methods' omission of vital physical information and their inability to incorporate iterative optimization details during inference. This oversight can compromise the quality of the solutions, particularly when navigating complex constraints and stringent performance requirements. Consequently, there's an emerging need for methodologies that amalgamate the strengths of both data-centric and physics-informed techniques to more adeptly navigate engineering applications. Paving the way forward, structured generative models have made notable strides in this domain, exemplified by innovative techniques like TopologyGAN [69] and TopoDiff [62].

**Limitations.** Structured generative models, despite their recent advancements in engineering designs, grapple with some pressing challenges. For one, these models often require additional supervised training data to learn guidance mechanisms that can improve performance and manufacturability [84]. In the context of Diffusion Models [43], executing forward simulations is not a one-off task; it requires repetition, often in the order of tens to hundreds of times, to derive an appropriate topology [62]. Moreover, integrating physical data into these models isn't straightforward. It demands a time-consuming FEA (Finite Element Analysis) preprocessing step during both the training and inference phases, which is pivotal for computing critical parameters like stress and energy fields[62, 69]. As a result, the sampling process is slow, and the inference process is computationally expensive. Such overheads challenge the scalability and adaptability of these models. Consequently, the touted benefits of data-driven techniques, notably fast sampling and quick design candidate generation, are somewhat diminished.

**Proposed Solution.** We introduce a conditional diffusion model that synergistically combines data-driven and optimization-based techniques. This model is tailored to learn constrained problems and generate candidates in the engineering design domains (refer to Fig. 3). Instead of relying on computationally heavy physics-based exact solutions using FEM, our method employs cost-effective physics-informed approximations to manage sparsity in conditioning constraints. We introduce a

Trajectory Alignment (TA) mechanism in the training phase that allows the model to leverage the information in the trajectory that was used by an iterative optimization-based method in the training data, drastically cutting down the sampling steps required for a solution generation using diffusion models. Moreover, our framework can amplify performance and manufacturability in complex problems by integrating a few steps of direct optimization. This method strikes a balance between computational efficiency and precision, offering adaptability to novel design challenges. By bridging the gap between generative modeling and engineering design optimization, our solution emerges as a powerful tool for tackling complex engineering problems. At its core, our findings highlight the value of diffusion models benefitting from optimization methods' intermediate solutions, emphasizing the journey and not just the final outcome.

**Contribution.** Our contributions are the following:

(i) We introduce the *Diffusion Optimization Models* (DOM), a versatile and efficient approach to incorporate performance awareness in generative models of engineering design problems while respecting constraints. The primary objective of DOM is to generate high-quality candidates rapidly and inexpensively, with a focus on topology optimization (TO) problems. DOM consists of

– *Trajectory Alignment* (TA) leverages iterative optimization and hierarchical sampling to match diffusion and optimization paths, distilling the optimizer knowledge in the sampling process. As a result, DOM achieves high performance without depending on FEM solvers or guidance and can sample high-quality configurations in as few as two steps.

– *Dense Kernel Relaxation*, an efficient mechanism to relieve inference from expensive FEM pre-processing and

– *Few-Steps Direct Optimization* that improves manufacturability using a few optimization steps.

(ii) We perform extensive quantitative and qualitative evaluation in- and out-of-distribution, showing how kernel relaxation and trajectory alignment are both necessary for good performance and fast, cheap sampling. We also release a large, *multi-fidelity dataset* of sub-optimal and optimal topologies obtained by solving minimum compliance optimization problems. This dataset contains low-resolution (64x64), high-resolution (256x256), optimal (120k), and suboptimal (600K) topologies. To our knowledge, this is the first large-scale dataset of optimized designs that also provides intermediate suboptimal iterations.

## 2   Background

Here we briefly introduce the Topology Optimization problem [14], diffusion models [105, 44, 98], a class of deep generative models, conditioning and guidance mechanisms for diffusion models, and deep generative models for topology optimization [69, 62]. For more related work see Appendix A.

**The Topology Optimization Problem.**   Topology optimization is a computational design approach that aims to determine the optimal arrangement of a structure, taking into account a set of constraints. Its objective is to identify the most efficient utilization of material while ensuring the structure meets specific performance requirements. One widely used method in topology optimization is the Solid Isotropic Material with Penalization (SIMP) method [13]. The SIMP method employs a density field to model the material properties, where the density indicates the proportion of material present in a

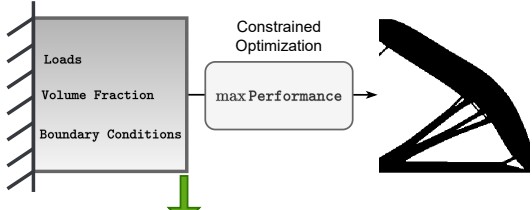

**Figure 2:** In topology optimization, the objective is to find the design with minimum compliance under given loads, boundary conditions, and volume fractions.

particular region. The optimization process involves iteratively adjusting the density field, considering constraints such as stress or deformation. In the context of a mechanical system, a common objective is to solve a generic minimum compliance problem. This problem aims to find the distribution of material density, represented as $\mathbf{x} \in \mathbb{R}^n$, that minimizes the deformation of the structure under prescribed boundary conditions and loads [59]. Given a set of design variables $\mathbf{x} = \{\mathbf{x}_i\}_{i=0}^n$, where

$n$ is the domain dimensionality, the minimum compliance problems can be written as:

$$\begin{aligned}
\min_{\mathbf{x}} \quad & c(\mathbf{x}) = F^T U(\mathbf{x}) \\
\text{s.t.} \quad & v(\mathbf{x}) = v^T \mathbf{x} < \bar{v} \\
& 0 \le \mathbf{x} \le 1
\end{aligned} \tag{1}$$

The goal is to find the design variables that minimize compliance $c(\mathbf{x})$ given the constraints. $F$ is the tensor of applied loads and $U(\mathbf{x})$ is the node displacement, solution of the equilibrium equation $K(\mathbf{x})U(\mathbf{x}) = F$ where $K(\mathbf{x})$ is the stiffness matrix and is a function of the considered material. $v(\mathbf{x})$ is the required volume fraction. The problem is a relaxation of the topology optimization task, where the design variables are continuous between 0 and 1. One significant advantage of topology optimization is its ability to create optimized structures that meet specific performance requirements. However, a major drawback of topology optimization is that it can be computationally intensive and may require significant computational resources. Additionally, some approaches to topology optimization may be limited in their ability to generate highly complex geometries and get stuck in local minima.

**Diffusion Models.** Let $\mathbf{x}_0$ denote the observed data $\mathbf{x}_0 \in \mathbb{R}^D$. Let $\mathbf{x}_1, ..., \mathbf{x}_T$ denote $T$ latent variables in $\mathbb{R}^D$. We now introduce, the *forward or diffusion process* $q$, the *reverse or generative process* $p_\theta$, and the objective $L$. The forward or diffusion process $q$ is defined as [44]: $q(\mathbf{x}_{1:T}|\mathbf{x}_0) = q(\mathbf{x}_1|\mathbf{x}_0) \prod_{t=2}^T q(\mathbf{x}_t|\mathbf{x}_{t-1})$. The beta schedule $\beta_1, \beta_2, ..., \beta_T$ is chosen such that the final latent image $\mathbf{x}_T$ is nearly Gaussian noise. The generative or inverse process $p_\theta$ is defined as: $p_\theta(\mathbf{x}_0, \mathbf{x}_{1:T}) = p_\theta(\mathbf{x}_0|\mathbf{x}_1)p(\mathbf{x}_T) \prod_{t=2}^T p_\theta(\mathbf{x}_{t-1}|\mathbf{x}_t)$. The neural network $\mu_\theta(\mathbf{x}_t, t)$ is shared among all time steps and is conditioned on $t$. The model is trained with a re-weighted version of the ELBO that relates to denoising score matching [105]. The negative ELBO $L$ can be written as:

$$\mathbb{E}_q \left[ -\log \frac{p_\theta(\mathbf{x}_0, \mathbf{x}_{1:T})}{q(\mathbf{x}_{1:T}|\mathbf{x}_0)} \right] = L_0 + \sum_{t=2}^T L_{t-1} + L_T, \tag{2}$$

where $L_0 = \mathbb{E}_{q(\mathbf{x}_1|\mathbf{x}_0)} \left[ -\log p(\mathbf{x}_0|\mathbf{x}_1) \right]$ is the likelihood term (parameterized by a discretized Gaussian distribution) and, if $\beta_1, ...\beta_T$ are fixed, $L_T = \mathbb{KL}[q(\mathbf{x}_T|\mathbf{x}_0), p(\mathbf{x}_T)]$ is a constant. The terms $L_{t-1}$ for $t = 2, ..., T$ can be written as: $L_{t-1} = \mathbb{E}_{q(\mathbf{x}_t|\mathbf{x}_0)} \left[ \mathbb{KL}[q(\mathbf{x}_{t-1}|\mathbf{x}_t, \mathbf{x}_0) \mid p(\mathbf{x}_{t-1}|\mathbf{x}_t)] \right]$. The terms $L_{1:T-1}$ can be rewritten as a prediction of the noise $\epsilon$ added to $\mathbf{x}$ in $q(\mathbf{x}_t|\mathbf{x}_0)$. Parameterizing $\mu_\theta$ using the noise prediction $\epsilon_\theta$, we can write

$$L_{t-1,\epsilon}(\mathbf{x}) = \mathbb{E}_{q(\epsilon)} \left[ w_t \|\epsilon_\theta(\mathbf{x}_t(\mathbf{x}_0, \epsilon)) - \epsilon\|_2^2 \right], \tag{3}$$

where $w_t = \frac{\beta_t^2}{2\sigma_t^2 \alpha_t (1-\bar{\alpha}_t)}$, which corresponds to the ELBO objective [50, 56].

**Conditioning and Guidance.** Conditional diffusion models have been adapted for constrained engineering problems with performance requirements. TopoDiff [62] proposes to condition on loads, volume fraction, and physical fields similarly to [69] to learn a constrained generative model. In particular, the generative model can be written as:

$$p_\theta(\mathbf{x}_{t-1}|\mathbf{x}_t, \mathbf{c}, \mathbf{g}) = \mathcal{N}(\mathbf{x}_{t-1}; \mu_\theta(\mathbf{x}_t, \mathbf{c}) + \sum_{p=1}^P \mathbf{g}_p, \ \gamma), \tag{4}$$

where $\mathbf{c}$ is a conditioning term and is a function of the loads $l$, volume fraction $v$, and fields $f$, i.e $\mathbf{c} = h(l, v, f)$. The fields considered are the Von Mises stress $\sigma_{vm} = \sqrt{\sigma_{11}^2 - \sigma_{11}\sigma_{22} + \sigma_{22}^2 + 3\sigma_{12}^2}$ and the strain energy density field $W = (\sigma_{11}\epsilon_{11} + \sigma_{22}\epsilon_{22} + 2\sigma_{12}\epsilon_{12})/2$. Here $\sigma_{ij}$ and $\epsilon_{ij}$ are the stress and energy components over the domain. $\mathbf{g}$ is a guidance term, containing information to guide the sampling process toward regions with low floating material (using a classifier and $\mathbf{g}_{fm}$) and regions with low compliance error, where the generated topologies are close to optimized one (using a regression model and $\mathbf{g}_c$). Where conditioning $\mathbf{c}$ is always present and applied during training, the guidance mechanism $\mathbf{g}$ is optional and applied only at inference time.

**TopoDiff Limitations.** TopoDiff is effective at generating topologies that fulfill the constraints and have low compliance errors. However, the generative model is expensive in terms of sampling time, because we need to sample tens or hundreds of layers for each sample. Additionally, given the model conditions on the Von Mises stress and the strain energy density, for each configuration of loads and

boundary conditions, we have to preprocess the given configurations running a FEM solver. This, other than being computationally expensive and time-consuming, relies on fine-grained knowledge of the problem at hand in terms of material property, domain, and input to the solver and performance metrics, limiting the applicability of such modeling techniques for different constrained problems in engineering or even more challenging topology problems. The guidance requires the training of two additional models (a classification and a regression model) and is particularly useful with out-of-distribution configurations. However such guidance requires additional topologies, optimal and suboptimal, to train the regression model, assuming that we have access to the desired performance metric on the train set. Similarly for the classifier, where additional labeled data has to be gathered.

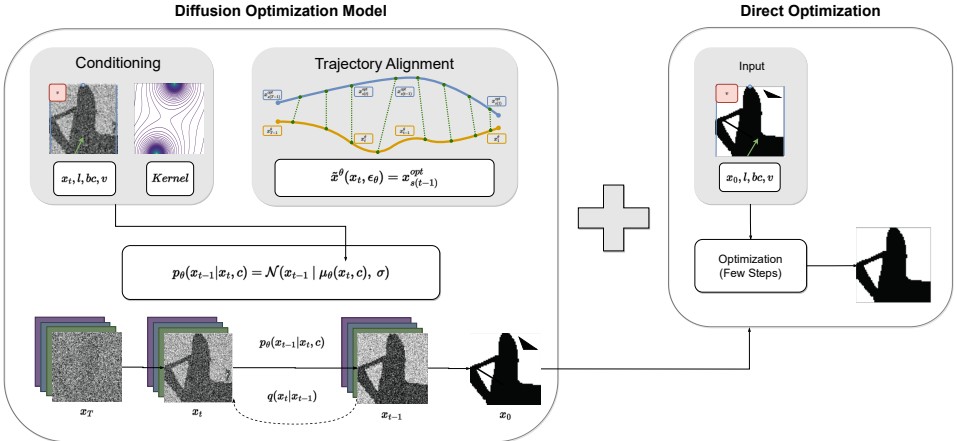

**Figure 3:** The DOM pipeline with conditioning and kernel relaxation (top left) and trajectory alignment (top right). The Diffusion Optimization Model generates design candidates, which are further refined using optimization tools. After the generation step (left side), we can improve the generated topology using a few steps of SIMP (5/10) to remove floating material and improve performance (right side). See Appendix 10 for a comparison of the inference process for DOM and TopoDiff [62].

## 3 Method

To tackle such limitations, we propose *Diffusion Optimization Models* (DOM), a conditional diffusion model to improve constrained design generation. One of our main goals is to improve inference time without loss in performance and constraint satisfaction. DOM is based on three main components: (i) Trajectory Alignment (Fig. 1 and Fig. 4) to ground sampling trajectory in the underlying physical process; (ii) Dense Kernel Relaxation (Fig. 5) to make preprocessing efficient; and (iii) Few-Steps Direct Optimization to improve out-of-distribution performance (Fig. 3 for an overview). See appendix C for algorithms with and without trajectory alignment.

**Empirical Methodology.** We would like to highlight that our primary contribution, trajectory alignment, is predominantly empirical. While we do make assumptions about the optimization and sampling trajectory and utilize TA, we have not established a comprehensive theoretical framework ensuring convergence of the regularized loss to the score optimized by a diffusion model. Nonetheless, our empirical findings provide evidence for the convergence of the sampling process to the desired solutions.

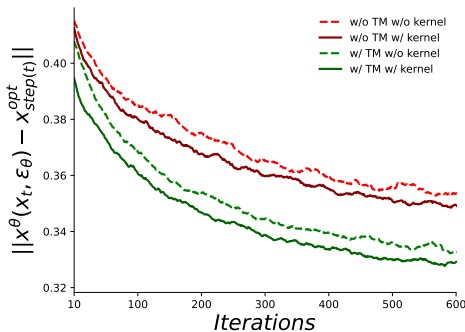

**Figure 4:** Distance between intermediate sampling steps in DOM and optimization steps with and without Trajectory Alignment. Given a random sampling step $t$ and the corresponding optimization step $s(t) = \mod(t, n)$ where $n \in [2, 10]$. We compute the matching in clean space, using the approximate posterior $q$ to obtain an estimate for $\mathbf{x}^\theta$ given $\mathbf{x}_t$ and the noise prediction $\epsilon_\theta$. Then we compute the distance $||\tilde{\mathbf{x}}^\theta(\mathbf{x}_t, \epsilon_\theta) - \mathbf{x}_{s(t)}^{opt}||_2$.

**Trajectory Alignment (TA).** Our goal is to align the sampling trajectory with the optimization trajectory, incorporating optimization in data-driven generative models by leveraging the hierarchical

sampling structure of diffusion models. This aligns trajectories with physics-based information, as illustrated in Fig. 1a. Unlike previous approaches, which use optimization as pre-processing or post-processing steps, trajectory alignment is performed during training and relies upon the marginalization property of diffusion models, i.e., $q(\mathbf{x}_t|\mathbf{x}_0) = \int q(\mathbf{x}_{1:t}|\mathbf{x}_0)d\mathbf{x}_{1:t-1}$, where $\mathbf{x}_t = \sqrt{\bar{\alpha}_t}\mathbf{x}_0 + (1-\bar{\alpha}_t)\,\epsilon$, with $\epsilon \sim N(0, I)$. The trajectory alignment process can match in clean space (matching step 0), noisy space (matching step $t$), performance space, and leverage multi-fidelity mechanisms. At a high level, TA is a regularization mechanism that injects an optimization-informed prior at each sampling step, forcing it to be close to the corresponding optimization step in terms of distance. This process provides a consistency mechanism [108, 104, 97] over trajectories and significantly reduces the computational cost of generating candidates without sacrificing accuracy.

**Alignment Challenges.** The alignment of sampling and optimization trajectories is challenging due to their differing lengths and structures. For example, the optimization trajectory starts with an image of all zeros, while the sampling path starts with random noise. Furthermore, Diffusion Models define a Stochastic Differential Equation (SDE, [106]) in the continuous limit, which represents a collection of trajectories, and the optimization trajectory cannot be directly represented within this set. To address these issues, trajectory alignment comprises two phases (see Figure 1b): a search phase and a matching phase. In the search phase, we aim to find the closest trajectory, among those that can be represented by the reverse process, to the optimization trajectory. This involves identifying a suitable representation over a trajectory that aligns with the optimization process.

In the matching phase, we minimize the distance between points on the sampling and optimization trajectories to ensure proximity between points and enable alignment between trajectories.

**Trajectory Search.** We leverage the approximate posterior and marginalization properties of diffusion models to perform a trajectory search, using the generative model as a parametric guide to search for a suitable representation for alignment. Given an initial point $\mathbf{x}_0$, we obtain an approximate point $\mathbf{x}_t$ by sampling from the posterior distribution $q(\mathbf{x}_t|\mathbf{x}_0)$. We then predict $\epsilon_\theta(\mathbf{x}_t)$ with the model and use it to obtain

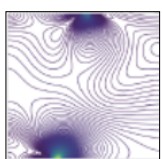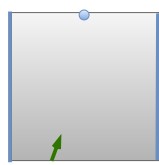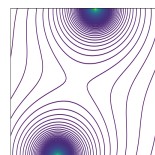

**Figure 5:** Comparison of iterative (left), sparse (center), and dense single-step (right) conditioning fields for a Constrained Diffusion Model. Unlike the expensive iterative FEA method, the physics-inspired fields offer a cost-effective, single-step approximation that's domain-agnostic and scalable.

$\tilde{\mathbf{x}}^\theta(\mathbf{x}_t, \epsilon_\theta(\mathbf{x}_t))$. In a DDPM, $\tilde{\mathbf{x}}^\theta$ is an approximation of $\mathbf{x}_0$ and is used as an intermediate step to sample $\mathbf{x}_{t-1}^\theta$ using the posterior functional form $q(\mathbf{x}_{t-1}^\theta|\mathbf{x}_t, \tilde{\mathbf{x}}^\theta)$. In DOM, we additionally leverage $\tilde{\mathbf{x}}^\theta$ to transport the sampling step towards a suitable representation for matching an intermediate optimization step $\mathbf{x}_{step(t)}^{opt}$ corresponding to $t$ using some mapping. Trajectory alignment involves matching the optimization trajectory, which is an iterative exact solution for physics-based problems, with the sampling trajectory, which is the hierarchical sampling mechanism leveraged in Diffusion Models [43] and Hierarchical VAEs [100]. In practice, in DOM we sample $\mathbf{x}_t = \sqrt{\bar{\alpha}_t}\mathbf{x}_0 + (1-\bar{\alpha}_t)\epsilon$ from $q(\mathbf{x}_t|\mathbf{x}_0)$ and run a forward step with the inverse process $\epsilon_\theta(\mathbf{x}_t, \mathbf{c})$ conditioning on the constraints $\mathbf{c}$ to obtain the matching representation $\tilde{\mathbf{x}}^\theta$ for step $t$:

$$\tilde{\mathbf{x}}^\theta \sim q(\tilde{\mathbf{x}}^\theta|\tilde{\mu}^\theta(\mathbf{x}_t, \epsilon_\theta), \gamma)$$
$$\tilde{\mu}^\theta(\mathbf{x}_t, \epsilon_\theta) = (\mathbf{x}_t - \sqrt{1-\bar{\alpha}_t}\,\epsilon_\theta(\mathbf{x}_t, \mathbf{c}))/\sqrt{\bar{\alpha}_t}. \tag{5}$$

**Trajectory Matching.** Then we match the distribution of matching representation $q(\tilde{\mathbf{x}}^\theta|\mathbf{x}_t, \epsilon_\theta)$ for sampling step $t$ with the distribution of optimized representations $q(\mathbf{x}_{s(t-1)}^{opt}|\texttt{opt})$ at iteration $s$ (corresponding to step $t-1$) conditioning on the optimizer $S$. In general, given that the sampling steps will be different than the optimization steps, we use $s(t) - 1 = n_s \times (1 - t/T)$, where $n_s$ is the number of optimized iterations stored, and then select the integer part. [1] We then can train the model as a weighted sum of the conditional DDPM objective and the trajectory alignment regularization:

$$\mathcal{L}_{\texttt{DOM}} = \mathbb{E}_{q(\mathbf{x}_t|\mathbf{x}_0)}\Big[\mathbb{KL}[q(\mathbf{x}_{t-1}|\mathbf{x}_t, \mathbf{x}_0)\,|\,p_\theta(\mathbf{x}_{t-1}^\theta|\mathbf{x}_t, \mathbf{c})] + \mathbb{KL}[q(\tilde{\mathbf{x}}^\theta|\mathbf{x}_t, \epsilon_\theta)\,|\,q(\mathbf{x}_{s(t-1)}|\texttt{opt})]\Big]. \tag{6}$$

This mechanism effectively pushes the sampling trajectory at each step to match the optimization trajectory, distilling the optimizer during the reverse process training. In practice, following practice

---

[1] for example, if $n_s = 5$ and $T = 1000$, $s = 1$ for $(T : T - 200)$, and $s = 5$ for steps $(200 : 1)$.

in DDPM literature, the distribution variances are not learned from data. For the trajectory alignment distributions, we set the dispersion to the same values used in the model. By doing so we can rewrite the per-step negated lower-bound as a weighted sum of squared errors:

$$\mathcal{L}_{\texttt{DOM}} = \underbrace{\mathbb{E}_{q(\epsilon)}\left[w_t\|\epsilon_\theta(\mathbf{x}_t(\mathbf{x}_0, \epsilon), \mathbf{c}) - \epsilon\|_2^2\right]}_{L_{t-1,\epsilon}(\mathbf{x}, \mathbf{c})} + \underbrace{\alpha_c\|\tilde{\mathbf{x}}^\theta(\mathbf{x}_t, \epsilon_\theta) - \mathbf{x}_{s(t-1)}^{opt}\|_2^2}_{\mathcal{L}_{clean}^{\texttt{TA}}} \tag{7}$$

where $\mathcal{L}_{clean}^{\texttt{TA}}$ is the trajectory alignment loss for step $t$, and $L_{t-1,\epsilon}(\mathbf{x}, \mathbf{c})$ is a conditional DDPM loss for step $t$. This is the formulation employed for our model, where we optimize this loss for the mean values, freeze the mean representations, and optimize the variances in a separate step [68]. Alignment can also be performed in alternative ways. We can perform matching in noisy spaces, using the marginal posterior to obtain a noisy optimized representation for step $t - 1$, $q(\mathbf{x}_{t-1}^{opt}|\mathbf{x}_{s(0)}^{opt})$ and then optimize $\mathcal{L}_{noisy}^{\texttt{TA}} = \alpha_n\|\mathbf{x}_{t-1}^\theta - \mathbf{x}_{t-1}^{opt}\|_2^2$. Finally, we can match in performance space: this approach leverages an auxiliary model $f_\phi$ similar to (consistency models) and performs trajectory alignment in functional space, $\mathcal{L}_{perf}^{\texttt{TA}} = \alpha_p\|f_\phi(\mathbf{x}_{t-1}^\theta) - P_{s(t-1)}\|_2$ , where we match the performance for the generated intermediate design with the ground truth intermediate performance $P_{s(t-1)}$ for the optimized $\mathbf{x}_{s(t-1)}^{opt}$. We compare these and other variants in Table 6.

**Dense Conditioning over Sparse Constraints.**    All models are subject to conditioning based on loads, boundary conditions, and volume fractions. In addition, TopoDiff and TopoDiff-GUIDED undergo conditioning based on force field and energy strain, while TopoDiff-FF and DOM are conditioned based on a dense kernel relaxation, inspired by Green's method [36, 34], which defines integral functions that are solutions to the time-invariant Poisson's Equation [33, 41]. More details are in Appendix D. The idea is to use the kernels as approximations to represent the effects of the boundary conditions and loads as smooth functions across the domain (Fig. 5). This approach avoids the need for computationally expensive and time-consuming Finite Element Analysis (FEA) to provide conditioning information. For a load or source $l$, a sink or boundary $b$ and $r = \|\mathbf{x} - \mathbf{x}_l\|_2 = \sqrt{(\mathbf{x}_i - \mathbf{x}_i^l)^2 + (\mathbf{x}_j - \mathbf{x}_j^l)^2}$, we have:

$$K_l(\mathbf{x}, \mathbf{x}_l; \alpha) = \sum_{l=1}^L (1 - e^{-\alpha/\|\mathbf{x}-\mathbf{x}_l\|_2^2})\, \bar{p}(\mathbf{x}_l)$$
$$K_b(\mathbf{x}, \mathbf{x}_b; \alpha) = \sum_{b=1}^B e^{-\alpha/\|\mathbf{x}-\mathbf{x}_b\|_2^2} / \max_{\mathbf{x}}\left(\sum_{b=1}^B e^{-\alpha/\|\mathbf{x}-\mathbf{x}_b\|_2^2}\right). \tag{8}$$

where $\bar{p}$ is the module of a generic force in 2D. Notice how, for $r \to 0$, $K_l(\mathbf{x}, \mathbf{x}_l) \to p$, and $r \to \infty$, $K_l(\mathbf{x}, \mathbf{x}_l) \to 0$. We notice how closer to the boundary the kernel is null, and farther from the boundary the kernel tends to 1. Note that the choice of $\alpha$ parameters in the kernels affects the smoothness and range of the kernel functions. Furthermore, these kernels are isotropic, meaning that they do not depend on the direction in which they are applied. Overall, the kernel relaxation method offers a computationally inexpensive way to condition generative models on boundary conditions and loads, making them more applicable in practical engineering and design contexts.

**Few-Steps Direct Optimization.**    Finally, we leverage direct optimization to improve the data-driven candidate generated by DOM. In particular, by running a few steps of optimization (5/10) we can inject physics information into the generated design directly, greatly increasing not only performance but greatly increasing manufacturability. Given a sample from the model $\tilde{\mathbf{x}}_0 \sim p_\theta(\mathbf{x}_0|\mathbf{x}_1)p_\theta(\mathbf{x}_{1:T})$, we can post-process it and obtain $\mathbf{x}_0 = \texttt{opt}(\tilde{\mathbf{x}}_0^\theta, n)$ an improved design leveraging $n$ steps of optimization, where $n \in [5, 10]$. In Fig. 3 we show a full pipeline for DOM.

## 4   Experiments

Our three main objectives are: **(1)** Improving inference efficiency, and reducing the sampling time for diffusion-based topology generation while still satisfying the design requirements with a minimum decrease in performance. **(2)** Minimizing reliance on force and strain fields as conditioning information, reducing the computation burden at inference time and the need for ad-hoc conditioning mechanisms for each problem. **(3)** Merging together learning-based and optimization-based methods, refining the topology generated using a conditional diffusion model, and improving the final solution in terms of manufacturability and performance.

**Setup.** We train all the models for 200k steps on 30k optimized topologies on a 64x64 domain. For each optimized topology, we have access to a small subset (5 steps) of intermediate optimization

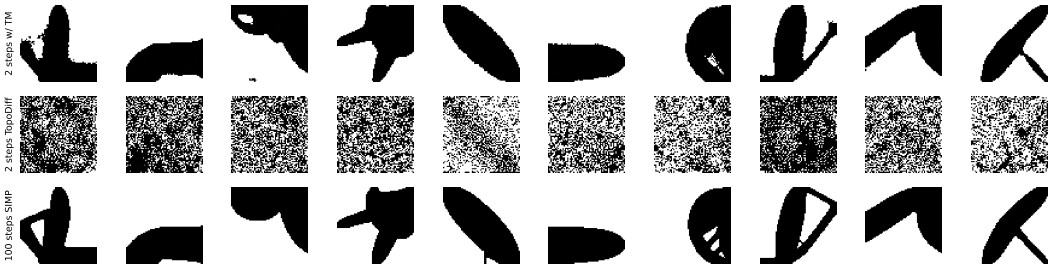

**Figure 6:** Few-Step sampling for Topology generation. Top row: Diffusion Optimization Model (DOM) with Trajectory Alignment. Middle row: TopoDiff-GUIDED. Bottom row: The optimization result. DOM produces high-quality designs in as few as two steps, greatly enhancing inference efficiency compared to previous models requiring 10-100 steps. Trajectory Alignment helps DOM generate near-optimal geometries swiftly, improving few-step sampling in conditional diffusion models for topology optimization. See appendix Fig. 13 for more examples.

steps. We set the hyperparameters, conditioning structure, and training routine as proposed in [62]. Appendix G for more details. For all the models (Table 1) we condition on volume fraction and loads. For TopoDiff, we condition additional stress and energy fields. For TopoDiff-FF [37], a variant of TopoDiff conditioning on a kernel relaxation, we condition on boundary conditions and kernels. TopoDiff-GUIDED leverages a compliance regressor and floating material classifier guidance. We use a reduced number of sampling steps for all the experiments.

**Dataset.** We use a dataset of optimized topologies gathered using SIMP as proposed in [66, 62]. Together with the topologies, the dataset contains information about optimal performance. For each topology, we have information about the loading condition, boundary condition, volume fraction, and optimal compliance. Additionally, for each constraint configuration, a pre-processing step computes the force and strain energy fields (see Fig. 5) when needed. Appendix F for more details on the dataset and visualizations.

**Evaluation.**

We evaluate the model using engineering and generative metrics. In particular, we consider metrics that evaluate how well our model fulfills: physical constraints using error wrt prescribed Volume Fraction (VFE); engineering constraints, as manufacturability as measured by Floating Material (FM); performance constraints, as measured by compliance error (CE) wrt the optimized SIMP solution; sampling time constraints (inference constraints) as measure by sampling time (inference and pre-processing). We consider two scenarios of increasing complexity: **(i)** In-distribution Constraints. The constraints in this test set are the same as those of the training set. When measuring performance on this set, we filter generated configurations with high compliance. **(ii)** Out-of-distribution Constraints. The constraints in this test set are different from those of

**Table 1:** Comparative study of generative models in topology optimization considering factors like conditional input (COND), finite element method (FEM), and guidance (GUID). Unlike others, the DOM model operates without FEM preprocessing or GUIDANCE. More visualizations and optimization trajectories are in the Appendix.

|  | w/ COND | w/o FEM | w/o GUID |
|---|---|---|---|
| TopologyGAN [69] | ✓ | ✗ | ✗ |
| TopoDiff [62] | ✓ | ✗ | ✓ |
| TopoDiff-G [62] | ✓ | ✗ | ✗ |
| DOM (ours) | ✓ | ✓ | ✓ |

the training set. When measuring performance on this set, we filter generated configurations with high compliance. The purpose of these tasks is to evaluate the generalization capability of the machine learning models in- and out-of-distribution. By testing the models on different test sets with varying levels of difficulty, we can assess how well the models can perform on new, unseen data. More importantly, we want to understand how important the role of the force field and energy strain is with unknown constraints.

**In-Distribution Constraints.** Table 3 reports the evaluation results in terms of constraints satisfaction and performance for the task of topology generation. In Table 2 we report metrics commonly employed to evaluate the quality of generative models in terms of fidelity (IS, sFID, P) and diversity (R). We see how such metrics are all close and it is challenging to gain any understanding just by relying on classic generative metrics when evaluating constrained design generation. These results justify the need for an evaluation that considers the performance and feasibility of the generated design.

In Table 3 DOM achieves high performance and is at least 50 % less computationally expensive at inference time, not requiring FEM preprocessing or additional guidance through surrogate models like TopologyGAN and TopoDiff. We also compare with Consistency Models [104], a Diffusion Model that tries to predict its input at each step. DOM can be seen as a generalization of such a method when a trajectory is available as a ground truth. Overall, DOM with Trajectory Alignment is competitive or better than the previous proposal in terms of performance on in-distribution constraints, providing strong evidence that TA is an effective mechanism to guide the sampling path toward regions of high performance.

**Table 2:** Generative metrics on in-distribution metrics. Precision denotes the fraction of generated topologies that are realistic, and recall measures the fraction of the training data manifold covered by the model.

| | IS ↑ | sFID ↓ | P ↑ | R ↑ |
|---|---|---|---|---|
| cDDPM | 3.41 | 40.30 | 0.73 | 0.85 |
| TopoDiff | 3.57 | 36.20 | **0.80** | **0.86** |
| TopoDiff-G | 3.63 | **35.96** | 0.79 | **0.86** |
| DOM w/o TA | 3.48 | 37.54 | 0.76 | **0.86** |
| DOM w/ TA | **3.68** | 36.73 | 0.77 | 0.85 |

**Table 3:** Evaluation of different model variants on in-distribution constraints. CE: Compliance Error. VFE: Volume Fraction Error. FM: Floating Material. We use 100 sampling steps for all diffusion models. We can see that DOM w/ TA is competitive with the SOTA on topology generation, being computationally 50 % less expensive at inference time compared to TopoDiff. Trajectory Alignment greatly improves performance without any additional inference cost. See appendix Fig. 11 for confidence intervals.

| | STEPS | CONSTRAINTS | AVG % CE ↓ | MDN % CE ↓ | % VFE ↓ | % FM ↓ | INFERENCE (s) ↓ |
|---|---|---|---|---|---|---|---|
| TopologyGAN [69] | 1 | FIELD | 48.51 | 2.06 | 11.87 | 46.78 | 3.37 |
| Conditional DDPM [68] | 100 | RAW | 60.79 | 3.15 | 1.72 | 8.72 | **2.23** |
| Consistency Model [104] | 100/1 | KERNEL | 10.30 | 2.20 | 1.64 | 8.72 | 2.35 |
| TopoDiff-FF [37] | 100 | KERNEL | 24.90 | 1.92 | 2.05 | 8.15 | 2.35 |
| TopoDiff [62] | 100 | FIELD | 5.46 | 0.80 | **1.47** | **5.79** | 5.54 |
| TopoDiff-GUIDED [62] | 100 | FIELD | 5.93 | 0.83 | 1.49 | 5.82 | 5.77 |
| DOM w/o TA (ours) | 100 | KERNEL | 13.61 | 1.79 | 1.86 | 7.44 | 2.35 |
| DOM w/ TA (ours) | 100 | KERNEL | **4.44** | **0.74** | 1.52 | 6.72 | 2.35 |

**Table 4:** Evaluating sampling topologies with few steps (2-10) for TopoDiff and DOM. G: Guided using regression and classifier guidance. AVG CE: average compliance error. MDN CE: median compliance error. VFE: volume fraction error. FM: floating material. INF: inference time. UNS: unsolvable configurations. LD: load disrespect. DOM largely outperforms TopoDiff in the few sampling step regimes, showing Trajectory Alignment's effectiveness as a grounding mechanism. DOM can generate reasonable topologies in just two sampling steps, where TopoDiff and DOM w/ TA fail completely, even presenting cases of load disrespect.

| *in-distro* | STEPS | SIZE | AVG % CE ↓ | MDN % CE ↓ | % VFE ↓ | % FM ↓ | INF (s) ↓ | % UNS ↓ | % LD ↓ |
|---|---|---|---|---|---|---|---|---|---|
| TopoDiff-G | 2 | 239M | 681.53 | 436.83 | 80.98 | 98.72 | 3.36 | **2.00** | 15.92 |
| DOM (ours) | 2 | 121M | **22.66** | **1.46** | **3.34** | **33.25** | **0.17** (- 94.94 %) | 2.11 | **0.00** |
| TopoDiff-G | 5 | 239M | 43.27 | 15.48 | 2.76 | 77.65 | 3.43 | **1.44** | 0.00 |
| DOM (ours) | 5 | 121M | **11.99** | **0.72** | **2.27** | **20.08** | **0.24** (- 93.00 %) | 2.77 | 0.00 |
| TopoDiff-G | 10 | 239M | 6.43 | 1.61 | 1.95 | 20.55 | 3.56 | **0.00** | 0.00 |
| DOM (ours) | 10 | 121M | **4.44** | **0.57** | **1.67** | **11.94** | **0.35** (- 90.17 %) | 0.00 | 0.00 |
| *out-distro* | | | | | | | | | |
| TopoDiff-G | 2 | 239M | 751.17 | 548.26 | 81.46 | 100.00 | 3.36 | **1.90** | 16.48 |
| DOM (ours) | 2 | 121M | **79.66** | **10.37** | **3.69** | **44.20** | **0.17** (- 94.94 %) | 2.80 | **0.00** |
| TopoDiff-G | 5 | 239M | 43.50 | 19.24 | 2.58 | 79.57 | 3.43 | 2.20 | 0.00 |
| DOM (ours) | 5 | 121M | **38.97** | **5.49** | **2.56** | **26.70** | **0.24** (- 93.00 %) | **1.40** | 0.00 |
| TopoDiff-G | 10 | 239M | **10.78** | **2.55** | 1.87 | 21.36 | 3.56 | 2.10 | 0.00 |
| DOM (ours) | 10 | 121M | 32.19 | 3.69 | **1.78** | **14.20** | **0.35** (- 90.17 %) | **0.40** | 0.00 |

**Generation with Few-Steps of Sampling.** Table 4 compares two different algorithms, TopoDiff-GUIDED and DOM, in terms of their performance when using only a few steps for sampling. The table shows the results of the in and out-of-distribution comparison, with TopoDiff-G and DOM both having STEPS values of 2, 5, and 10, and SIZE of 239M and 121M. We can see that DOM outperforms by a large margin TopoDiff-G when tasked with generating a new topology given a few steps, corroborating our hypothesis that aligning the sampling and optimization trajectory is an effective mechanism to obtain efficient generative models that satisfy constraints. DOM outperforms

TopoDiff-GUIDED even being 50 % smaller, without leveraging an expensive FEM solver for conditioning but relying on cheap dense relaxations, making it 20/10 times faster at sampling, and greatly enhancing the quality of the generated designs, providing evidence that Trajectory Alignment is an effective mechanism to distill information from the optimization path. In Fig. 6, we provide qualitative results to show how DOM (top row) is able to generate reasonable topologies, resembling the fully optimized structure running SIMP for 100 steps (bottom row), with just two steps at inference time, where the same model without TA or a TopoDiff are not able to perform such task. Overall these results corroborate our thesis regarding the usefulness of trajectory alignment for high-quality constrained generation. For more experiments on inference see Appendix B.

**Merging Generative Models and Optimization for Out-of-Distribution Constraints.** Table 5 shows the results of experiments on out-of-distribution constraints. In this scenario, employing FEM and guidance significantly enhances the performance of TopoDiff. Conditioning on the FEM output during inference can be seen as a form of test-time conditioning that can be adapted to the sample at hand. However, merging DOM and a few iterations of optimization is extremely effective in solving this problem, in particular in terms of improving volume fraction and floating material. Using the combination of DOM and SIMP is a way to impose performance constraints in the model without the need for surrogate models or guidance.

**Table 5:** Out-of-Distribution Scenario Comparison: TopoDiff-G outperforms DOM due to its adaptive conditioning mechanism, which leverages expensive FEM-computed fields. However, DOM coupled with a few steps of direct optimization (5/10) greatly surpasses TopoDiff in performance and manufacturability. This underscores the effectiveness of integrating data-driven and optimization methods in constrained design creation.

| | STEPS | MDN % CE ↓ | % VFE ↓ | % FM ↓ |
|---|---|---|---|---|
| TopoDiff-FF | 100 | 16.06 | 1.97 | 8.38 |
| TopoDiff-G | 100 | 1.82 | 1.80 | 6.21 |
| DOM | 100 | 3.47 | 1.59 | 8.02 |
| DOM + SIMP | 100+5 | 1.89 | 1.77 | 10.19 |
| DOM + SIMP | 100+10 | **1.15** | **1.10** | **2.61** |

**Trajectory Alignment Ablation.** The core contribution of DOM is trajectory alignment, a method to match sampling and optimization trajectories of arbitrary length and structure mapping intermediate steps to appropriate CLEAN (noise free or with reduced noise using the model and the marginalization properties of DDPM) representations. However, alignment can be performed in multiple ways, leveraging NOISY representation, matching performance (PERF), and using data at a higher resolution to impose consistency (MULTI). In Table 6 we perform an ablation study, considering DOM with and without kernel relaxation, and leveraging different kinds of trajectory matching. From the

**Table 6:** Ablation study with and without kernel and trajectory alignment. We explore different ways to match the sampling and optimization trajectory and we measure the Median Compliance Error. TA: trajectory alignment. CM: Consistency Models [104].

| | KERNEL | TA | MODE | IN-DISTRO | OUT-DISTRO |
|---|---|---|---|---|---|
| DOM | ✗ | ✗ | - | 3.29 | 8.05 |
| DOM | ✗ | ✓ | CLEAN | 1.11 | 9.01 |
| CM | ✓ | ✗ | - | 2.20 | 5.25 |
| DOM | ✓ | ✗ | - | 1.80 | 5.62 |
| DOM | ✓ | ✓ | MULTI | 34.95 | 54.73 |
| DOM | ✓ | ✓ | NOISY | 2.08 | 6.23 |
| DOM | ✓ | ✓ | PERF | 2.41 | 6.82 |
| DOM | ✓ | ✓ | CLEAN | **0.74** | **3.47** |

table, we see that using dense conditioning is extremely important for out-of-distribution performance, and that matching using CLEAN is the most effective method in and out-of-distribution. In Fig. 4 we report a visualization of the distance between sampling and optimization trajectory during training. From this plot, we can see how the kernel together with TA helps the model to find trajectories that are closer to the optimal one, again corroborating the need for dense conditioning and consistency regularization.

## 5   Conclusion

We presented Diffusion Optimization Models, a generative framework to align the sampling trajectory with the underlying physical process and learn an efficient and expressive generative model for constrained engineering design. Our work opens new avenues for improving generative design in engineering and related fields. However, our method is limited by the capacity to store and retrieve intermediate optimization steps, and, without a few steps of optimization, it underperforms out-of-distribution compared to FEM-conditional and guided methods.

## Acknowledgment

We would like to thank Noah Joseph Bagazinski, Kristen Marie Edwards, Amin Heyrani Nobari, Cyril Picard, Lyle Regenwetter and Binyang Song for insightful comments and useful discussions. We extend our gratitude to the anonymous reviewers for their constructive feedback and discussions that greatly enhanced the clarity and quality of the paper. OW's work was funded in part by the Novo Nordisk Foundation through the Center for Basic Machine Learning Research in Life Science (NNF20OC0062606) and by the Pioneer Centre for AI, DNRF grant number P1.

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

# A  Related Work

**Topology Optimization.**  Engineering design is the process of creating solutions to technical problems under engineering requirements [22, 91, 23]. Often, the goal is to create highly performative designs given the required constraints. Topology Optimization (TO [14]) is a branch of engineering design and is a critical component of the design process in many industries, including aerospace, automotive, manufacturing, and software development. From the inception of the homogenization method for TO, a number of different approaches have been proposed, including density-based [13, 86, 64], level-set [3, 111], derivative-based [99], evolutionary [115], and others [19]. The density-based methods are widely used and use a nodal-based representation where the level-set leverages shape derivative to obtain the optimal topology. To improve the final design, filtering mechanisms have been proposed [117, 39, 94]. Hybrid methods are also widely used. Topology Optimization has evolved as a more and more intensive computational discipline, with the availability of efficient open-source implementations [92, 93, 46, 59, 4]. See [59] for more on this topic. See [95, 96] for a comprehensive review of the Topology Optimization field.

**Generative Models for Topology Optimization.**  Following the success of Deep Learning (DL) in vision, a surging interest arose recently for transferring these methods to the engineering field. In particular, DL methods have been employed for direct-design [1, 5, 11, 60, 110], accelating the optimization process [7, 49, 52, 107, 118], improving the shape optimization post-processing [42, 119], super-resolusion [32, 65, 120], sensitivity analysis [6, 75, 10, 89], 3d topologies [54, 87, 11], and more [26, 24, 25, 29]. Among these methods, Generative Models are especially appealing to improve design diversity in engineering design [2, 72, 70, 71]. In TO, the work of [47, 82, 83, 55, 109] focus on increasing diversity leveraging data-driven approaches. Additionally, Generative Models have been used for Topology Optimization problems conditioning on constraints (loads, boundary conditions, volume fraction for the structural case), directly generating topologies [81, 90, 40] training dataset of optimized topologies, leveraging superresolution methods to improve fidelity [121, 58], using filtering and iterative design approaches [74, 73, 17, 35] to improve quality and diversity. Methods for 3D topologies have also been proposed [12, 55]. Recently, GAN [69] and DDPM-based [62] approaches, conditioning on constraints and physical information, have had success in modeling the TO problem. For a comprehensive review and critique of the field, see [113].

**Conditional Diffusion Models.**  Methods to condition DDPM have been proposed, conditioning at sampling time [28], learning a class-conditional score [106], explicitly conditioning on class information [68], features [38, 9], and physical properties [116, 45]. Recently, TopoDiff [62] has shown that conditional diffusion models with guidance [31] are effective for generating topologies that fulfill the constraints and have high manufacturability and high performance. TopoDiff relies on physics information and surrogate models to guide the sampling of novel topologies with good performance. Alternatives to speed up sampling in TopoDiff have been recently proposed [37], trading performance for fast candidate generation. Improving efficiency and sampling speed for diffusion models is an active research topic, both reducing the number of sampling steps [68, 103, 102, 8], improving the ODE solver [48, 112, 57], leveraging distillation [88, 63], and exploiting autoencoders for dimensionality reduction [85, 77, 76]. Reducing the number of sampling steps can also be achieved by improving the property of the hierarchical latent space, exploiting a form of consistency regularization [108, 97, 122] during training. Consistency Models [104] proposes reconstructing its input from any step in the diffusion chain, effectively forcing the model to reduce the sampling steps needed for high-quality sampling. We similarly want to improve the latent space properties but leverage trajectory alignment with a physical process. Recently, energy-constrained diffusion models [114] have been proposed to regularize graph learning a learn expressive representations for structured data.

**Code:** https://github.com/georgosgeorgos/trajectory-alignment-diffusion

# B  Additional Experiments

## B.1  Sampling Topologies in Few-Steps

**Table 7:** Evaluating sampling topologies with few steps (2-10) for TopoDiff and DOM. G: Guided using regression and classifier guidance. AVG CE: average compliance error. MDN CE: median compliance error. VFE: volume fraction error. FM: floating material. INF: inference time. UNS: unsolvable configurations. LD: load disrespect.

| | STEPS | SIZE | AVG % CE ↓ | MDN % CE ↓ | % VFE ↓ | % FM ↓ | INF (s) ↓ | % UNS ↓ | % LD ↓ |
|---|---|---|---|---|---|---|---|---|---|
| *in-distro* | | | | | | | | | |
| TopoDiff-G | 2 | 239M | 681.53 | 436.83 | 80.98 | 98.72 | 3.36 | **2.00** | 15.92 |
| DOM (ours) | 2 | 121M | **22.66** | **1.46** | **3.34** | **33.25** | **0.17** (- 94.94 %) | 2.11 | **0.00** |
| TopoDiff-G | 5 | 239M | 43.27 | 15.48 | 2.76 | 77.65 | 3.43 | **1.44** | 0.00 |
| DOM (ours) | 5 | 121M | **11.99** | **0.72** | **2.27** | **20.08** | **0.24** (- 93.00 %) | 2.77 | **0.00** |
| TopoDiff-G | 10 | 239M | 6.43 | 1.61 | 1.95 | 20.55 | 3.56 | **0.00** | **0.00** |
| DOM (ours) | 10 | 121M | **4.44** | **0.57** | **1.67** | **11.94** | **0.35** (- 90.17 %) | **0.00** | **0.00** |
| *out-distro* | | | | | | | | | |
| TopoDiff-G | 2 | 239M | 751.17 | 548.26 | 81.46 | 100.00 | 3.36 | **1.90** | 16.48 |
| DOM (ours) | 2 | 121M | **79.66** | **10.37** | **3.69** | **44.20** | **0.17** (- 94.94 %) | 2.80 | **0.00** |
| TopoDiff-G | 5 | 239M | 43.50 | 19.24 | 2.58 | 79.57 | 3.43 | 2.20 | 0.00 |
| DOM (ours) | 5 | 121M | **38.97** | **5.49** | **2.56** | **26.70** | **0.24** (- 93.00 %) | 1.40 | **0.00** |
| TopoDiff-G | 10 | 239M | **10.78** | **2.55** | 1.87 | 21.36 | 3.56 | 2.10 | 0.00 |
| DOM (ours) | 10 | 121M | 32.19 | 3.69 | **1.78** | **14.20** | **0.35** (- 90.17 %) | 0.40 | **0.00** |

## B.2  Inference Time

With the previous experiments, we proved that a data-driven approach, biased towards the physical process, can distill the optimization process and sample a novel topology in a few steps. We then provide experiments for in and out-of-distribution and ablate our choice of kernel relaxation and trajectory alignment mechanism. However, the final goal of data-driven design is to learn a general tool for fast candidate generation. In Table 8 we compare inference time for different models and, more importantly, for low- (64) and high-resolution (256). Given the computational burden of training such models at high resolution, we train all the models for only 10k steps (around 5% of the full training), and then we use them for sampling. With this experiment, we want to emphasize how fast DOM can perform inference compared to a SOTA model. We choose to run all the generative models and optimized for 100 steps: this setting is not suited for DOM, because as we have seen the model excels in the few-step sampling task. Table 8 presents a comparison of various models based on different factors such as resolution (RES), size, preprocess time, postprocess time, sampling, and inference time.

**Table 8:** Inference time for different models at low and high resolution. For all the diffusion models we sample 100 steps and for SIMP we iterate for 100 steps. *We report optimization time for a full comparison, but it is important to emphasize that SIMP runs on CPU and the DDPM-based models on GPU. $\Delta T_{inference} = (T_{model} - T_{topodiff})/T_{topodiff}$

| | RES | SIZE | PREPROCESS | POSTPROCESS | SAMPLING | INFERENCE | $\Delta T_{inference}$ |
|---|---|---|---|---|---|---|---|
| TopoDiff | 64 | 121M | 3.31 | 0.00 | 2.23 | 5.54 | + 0.00 % |
| TopoDiff-GUIDED | 64 | 239M | 3.31 | 0.00 | 2.46 | 5.77 | + 4.15 % |
| DOM (ours) | 64 | 121M | 0.12 | 0.00 | 2.23 | **2.35** | - 57.58 % |
| SIMP | 64 | - | 0.00 | 18.12 | 0.00 | 18.12 | + 227.07 %* |
| TopoDiff | 256 | 553M | 31.84 | 0.00 | 7.78 | 38.62 | + 0.00 % |
| TopoDiff-GUIDED | 256 | 1092M | 31.84 | 0.00 | 8.46 | 40.30 | + 4.35 % |
| DOM (ours) | 256 | 553M | 0.31 | 0.00 | 7.78 | **8.09** | - 79.05 % |
| SIMP | 256 | - | 0.00 | 316.02 | 0.00 | 316.02 | + 718.28 %* |

# C Algorithms

---

**Algorithm 1** `DOM with Trajectory Alignment`

---

**Require:** Optimized Topologies $\mathbf{X}_0$
**Require:** Constraints $C = (BC, L, VF)$
**Require:** Intermediate Optimization Steps $\mathbf{X}^{opt}$
  **while** Training **do**
     Sample batch $(\mathbf{x}_0, c, \mathbf{x}^{opt})$
     Compute Dense Relaxation $k = K(bc, l)$
     Compute Conditioning $\mathbf{c} = (k, c)$
     Sample $t, \epsilon, \mathbf{x}^{opt}_{s(t)}$
     Compute $\mathbf{x}_t \sim q(\mathbf{x}_t | \mathbf{x}_0)$
     Forward Model $\epsilon_\theta(\mathbf{x}_t, \mathbf{c})$
     Compute Loss $L_{t-1}(\mathbf{x}, \mathbf{c}) = ||\epsilon_\theta(\mathbf{x}_t, \mathbf{c}) - \epsilon||_2^2$
     Trajectory Search $\tilde{\mathbf{x}}^\theta(\mathbf{x}_t, \epsilon_\theta) = (\mathbf{x}_t - \sqrt{1 - \bar{\alpha}_t}\, \epsilon_\theta(\mathbf{x}_t, \mathbf{c}))/\sqrt{\bar{\alpha}_t}$
     Trajectory Matching $\mathcal{L}_{\text{TA}} = ||\tilde{\mathbf{x}}^\theta(\mathbf{x}_t, \epsilon_\theta) - \mathbf{x}^{opt}_{s(t)}||_2^2$
     Compute Loss $\mathcal{L}_{\text{DOM}}(\theta) = L_{t-1}(\mathbf{x}, \mathbf{c}) + \mathcal{L}_{\text{TA}}$
     Backpropagate $\theta \leftarrow \nabla_\theta \mathcal{L}_{\text{DOM}}(\theta)$
  **end while**

---

**Algorithm 2** `DOM without Trajectory Alignment`

---

**Require:** Optimized Topologies $\mathbf{X}_0$
**Require:** Constraints $C = (BC, L, VF)$
  **while** Training **do**
     Sample batch $(\mathbf{x}_0, c)$
     Compute Dense Relaxation $k = K(bc, l)$
     Compute Conditioning $\mathbf{c} = (k, c)$
     Sample $t, \epsilon$
     Compute $\mathbf{x}_t \sim q(\mathbf{x}_t | \mathbf{x}_0)$
     Forward Model $\epsilon_\theta(\mathbf{x}_t, \mathbf{c})$
     Compute Loss $L_{t-1}(\mathbf{x}, \mathbf{c}) = ||\epsilon_\theta(\mathbf{x}_t, \mathbf{c}) - \epsilon||_2^2$
     Compute Loss $\mathcal{L}_{\text{DOM}}(\theta) = L_{t-1}(\mathbf{x}, \mathbf{c})$
     Backpropagate $\theta \leftarrow \nabla_\theta \mathcal{L}_{\text{DOM}}(\theta)$
  **end while**

---

```python
import numpy as np
import torch as th

def compute_kernel_load(batch_load_sample, axis):

    size = batch_load_sample.size(-1)
    if axis == "x":
        ix = 0
        xx = th.argwhere(batch_load_sample[0] != 0)
        coord = xx
    elif axis == "y":
        ix = 1
        yy = th.argwhere(batch_load_sample[1] != 0)
        coord = yy

    if len(coord) == 0:
        return batch_load_sample[ix], []

    x_grid = th.tensor([i for i in range(size)])
    y_grid = th.tensor([j for j in range(size)])

    kernel_load = 0
    for l in range(len(coord)):
        x_grid = th.tensor([i for i in range(size)])
        y_grid = th.tensor([j for j in range(size)])
        # distance
        x_grid = x_grid - coord[l][0]
        y_grid = y_grid - coord[l][1]

        grid = th.meshgrid(x_grid, y_grid)

        r_load = th.sqrt(grid[0]**2 + grid[1]**2)

        if axis == "x":
            p = batch_load_sample[0][coord[l][0], coord[l][1]]
        elif axis == "y":
            p = batch_load_sample[1][coord[l][0], coord[l][1]]

        kernel = 1 - th.exp(- 1/r_load**2)
        kernel_load += kernel * p

    return kernel_load, coord
```

**Listing 1:** Dense Kernel Relaxation for Sparse Loads.

# D   Physics-based Conditioning on Constraints

Seeking a more efficient diffusion-based topology generation while reducing dependency on force and strain fields, our approach is to approximate boundary conditions and loads using kernels. These kernels approximate the impact of constraints on the domain. The kernel architecture we opt for draws inspiration from Green's method [36, 34, 16, 41]. This method establishes integral functions as solutions to the time-invariantPoisson's Equation [33, 41], an extended form of Laplace's Equation addressing point source excitations. Expressed mathematically, Poisson's Equation is given as $\nabla_x^2 f(x) = h$, with $h$ as the force term and $f$ representing a general function over domain $\mathcal{X}$. This equation underpins numerous natural phenomena, with a notable case being a force part $h = 0$. This particular case brings forth Laplace's Equation, which is frequently used in heat transfer scenarios.

Green's method, part of a larger family of methods, offers a structured approach to address partial differential equations, even when domain specifics are unknown. The solutions derived using this method are termed Green's functions [53]. Even though such solutions can be generally complex, for a broad range of physical issues where constraints and forces are point-approximated, a straightforward functional framework can be established, hinging on the source and sink concept.

Consider, for instance, a laminar domain like a beam or a plate, restrained in a feasible manner. If a point force is applied (such as a downward pressure on a beam's edge or a plate's center) at $x_f$, this can be captured using the Dirac delta function, $\delta(x - x_f)$. The delta function is highly discontinuous but has powerful integration properties, specifically $\int f(x)\delta(x - x_f)dx = f(x_f)$ within domain $\mathcal{X}$. The solution for the time-invariant Poisson's Equation, when focusing on pinpoint forces, can be represented as a Green's function solution. This solution is primarily influenced by the distance from where the force is applied. Specifically:

$$\mathcal{G}(x, x') = -\frac{1}{4\pi} \frac{1}{|x - x'|}, \tag{9}$$

where $r = |x - x'| = \sqrt{|x_i - x_i'|^2 + |x_j - x_j'|^2}$. To approximate the forces and loads impacting our topologies, we utilize a kernel approximation grounded in Green's functions. While this model may not precisely map to all loads and boundary conditions, it does furnish us with computationally efficient conditioning data that aligns with core physical and engineering constraints. Harnessing these concepts, we aspire to enrich the conditioning data for the model, aiming to elevate generative models constrained by such conditions.

**Table 9:** Design and Modelling requirements for a constrained generative model for topology optimization. Our goal is to improve the requirements that are challenging to fulfill. In this work, we focus on improving Floating Material, reducing Compliance Error, and reducing Sampling Time.

| CLASS | METRICS | GOAL |
|---|---|---|
| Hard-constraint | Loads Disrespect | Feasibility |
| Hard-constraint | Floating Material | Manufacturability |
| Soft-constraint | Volume Fraction | Min Cost |
| Functional Performance | Compliance Error | Max Performance |
| Modeling Requirements | Sampling Time | Fast Inference |

# E   Tables and Visualizations

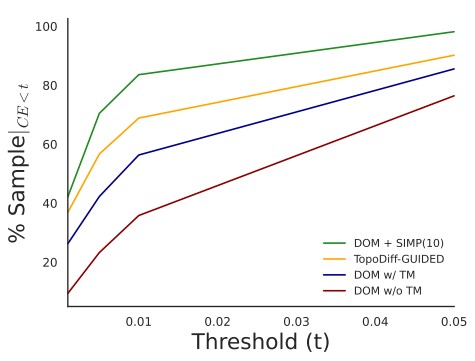

**(a)** Ratio of generated samples with CE $< t$.

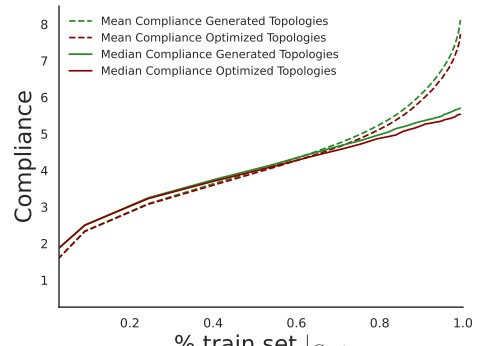

**(b)** Ratio of the training set with compliance $< t$.

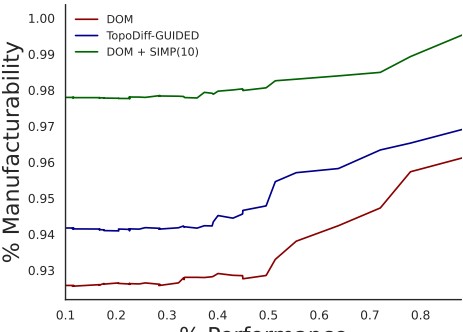

**(c)** Manufacturability vs Performance.

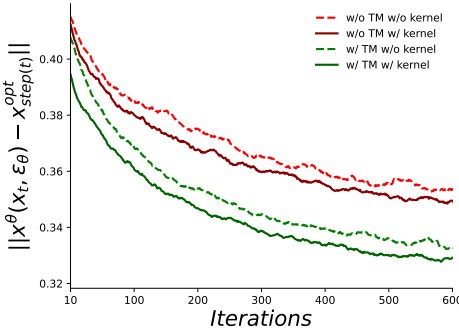

**(d)** Distance sampling and optimization path.

**Table 10:** Conditioning or guiding variables for different optimization methods and model configurations.

| | Load | BC | Kernel Load | Kernel BC | Force Field | Energy Field | VF | Performance |
|---|---|---|---|---|---|---|---|---|
| SIMP [14] | ✓ | ✓ | ✗ | ✗ | ✗ | ✗ | ✓ | ✓ |
| TopologyGAN [69] | ✓ | ✓ | ✗ | ✗ | ✓ | ✓ | ✓ | ✗ |
| TopoDiff [62] | ✓ | ✓ | ✗ | ✗ | ✓ | ✓ | ✓ | ✓ |
| TopoDiff-FF [37] | ✓ | ✓ | ✓ | ✓ | ✗ | ✗ | ✓ | ✗ |
| DOM (ours) | ✓ | ✓ | ✓ | ✓ | ✗ | ✗ | ✓ | ✗ |
| DOM + Trajectory Alignment (ours) | ✓ | ✓ | ✓ | ✓ | ✗ | ✗ | ✓ | ✓ |
| DOM + Optimizer (ours) | ✓ | ✓ | ✓ | ✓ | ✗ | ✗ | ✓ | ✓ |

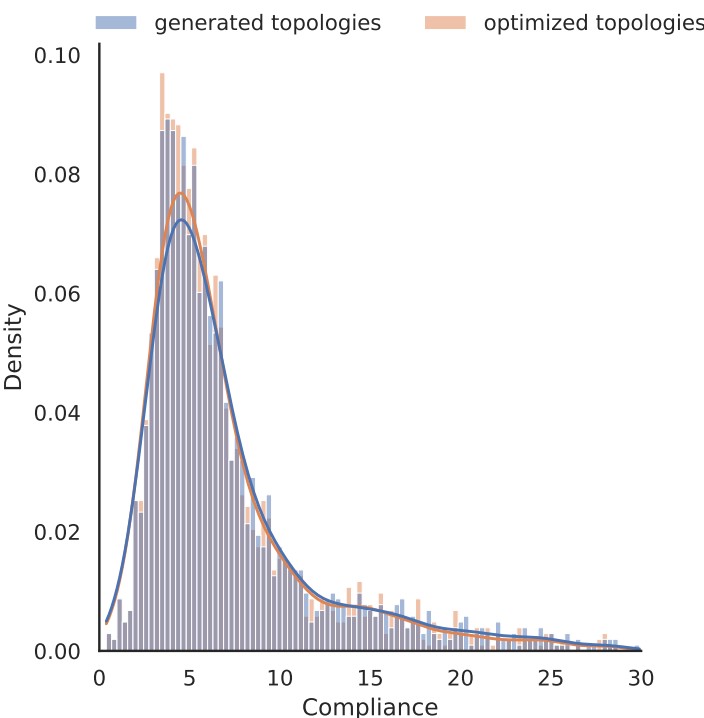

**Figure 8:** Histogram empirical distribution compliance for generated and optimized topologies.

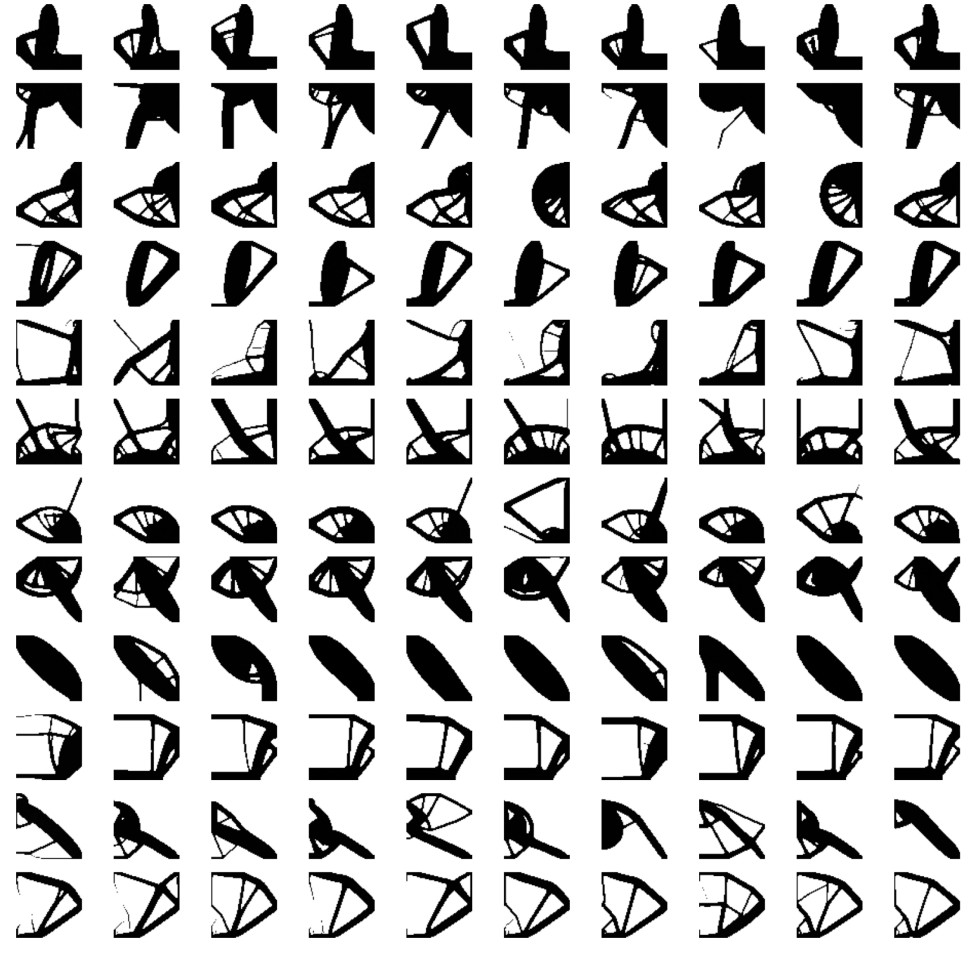

**Figure 9: DOM Sampling**. Conditional Samples from a Diffusion Optimization Model trained with Trajectory Alignment. For each row, we select a constraint configuration (boundary condition, loads, volume fraction) and sample the model 10 times. We use 100 sampling steps. We repeat this process 12 times. During sampling, DOM is a standard conditional DDPM and does not have access to the optimization trajectory.

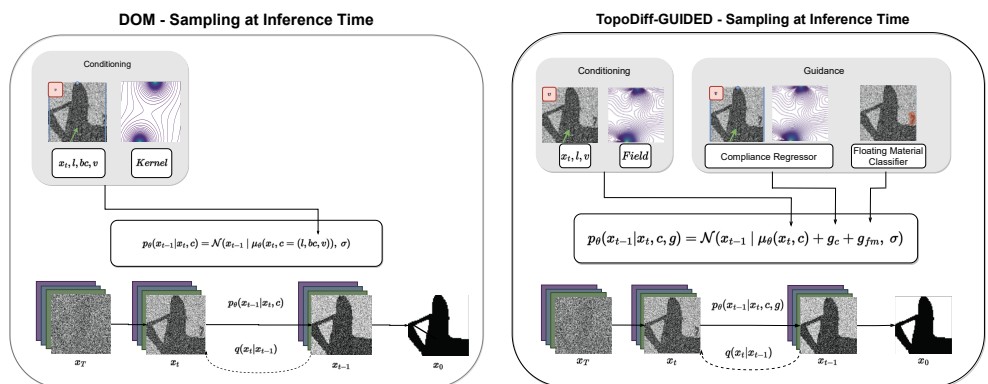

**Figure 10: DOM Sampling Process at Inference Time**. DOM (left) and TopoDiff-Guided (right) sampling process during inference. DOM is a conditional diffusion model. DOM is conditioned on the boundary conditions, loads, volume fraction, and kernels. After conditioning, we sample DOM with standard ancestral sampling like any DDPM. Contrary to TopoDiff (right), DOM does not rely on expensive preprocessing to compute energy and strain fields and auxiliary models for guidance. DOM can generate samples in a few steps (2/5) contrary to TopoDiff-G which requires hundreds of steps to generate reasonable topologies.

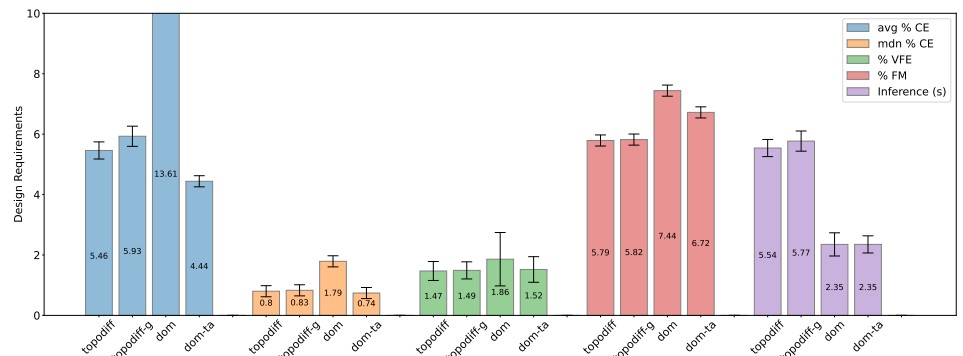

**Figure 11:** Confidence interval for design requirements on in-distribution constraint configurations.

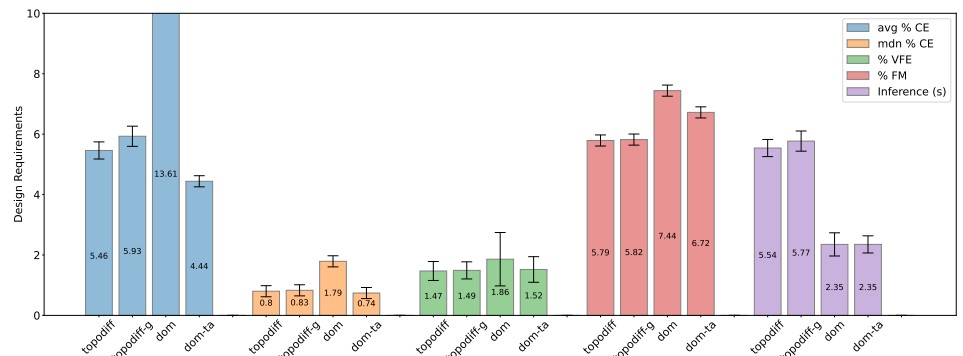

**Figure 12:** Examples of generated topologies with good performance.

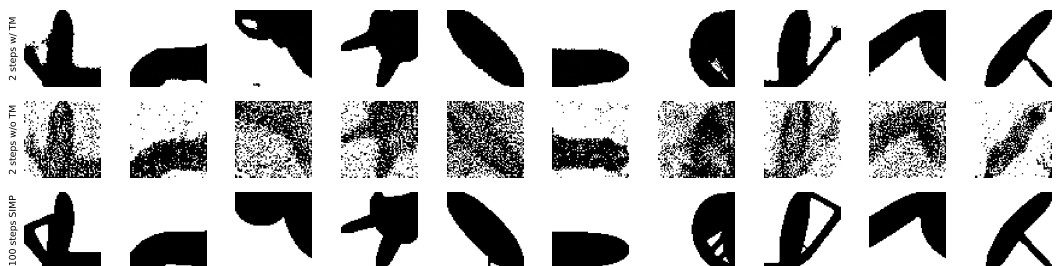

**Figure 13:** Comparison DOM w/ TA and DOM w/o TA generation with 2 steps.

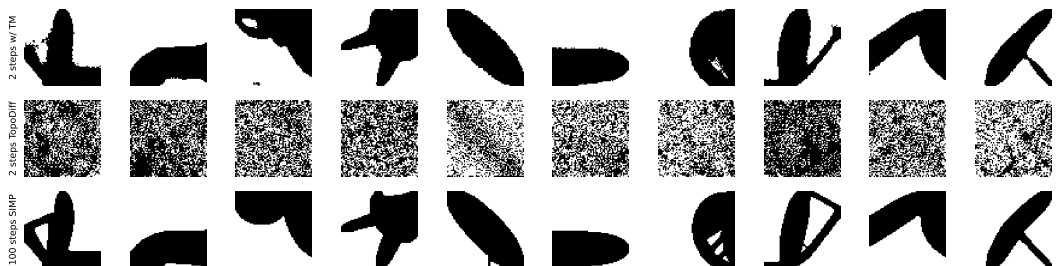

**Figure 14:** Comparison DOM w/ TA and TopoDiff-GUIDED generation with 2 steps.

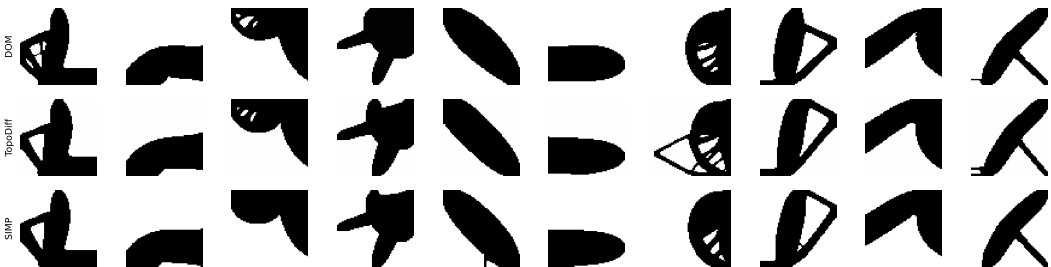

**Figure 15:** Comparison DOM and TopoDiff-GUIDED generation with 100 steps.

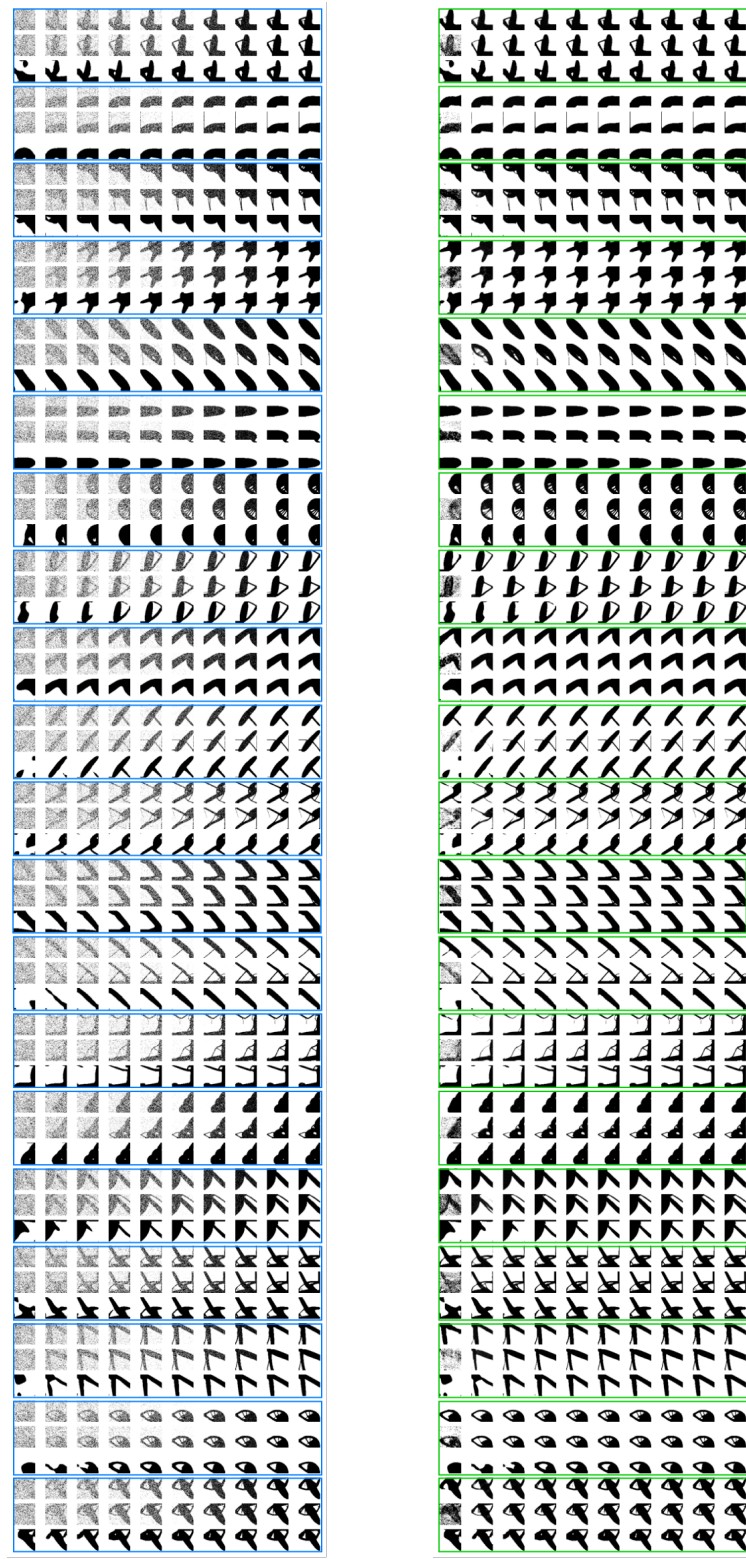

**Figure 16:** Left: $\mathbf{x}_t^\theta$ from 10 intermediate generation steps for DOM w/ TA (top block), DOM w/o TA (middle block), SIMP iterations (bottom block). Right: Prediction of $\tilde{\mathbf{x}}^\theta$ from 10 intermediate generation steps for DOM w/ TA (top block), DOM w/o TA (middle block), SIMP iterations (bottom block).

# F Dataset

We build a dataset of optimized topologies and intermediate optimization steps at low-resolution (64x64) and high-resolution (256x256). In particular:

- 50K low-resolution optimized topologies w/ constraints.
- 60K high-resolution optimizer topologies w/ constraints.
- 250K low-resolution intermediate steps [10, 20, 30, 50, 70] w/ constraints.
- 300K high-resolution intermediate steps [10, 20, 30, 50, 70] w/ constraints.

In Figure 17, 18, 19, 20 we show examples of intermediate steps at 10, 20, 30, and optimized topologies at high-resolution.

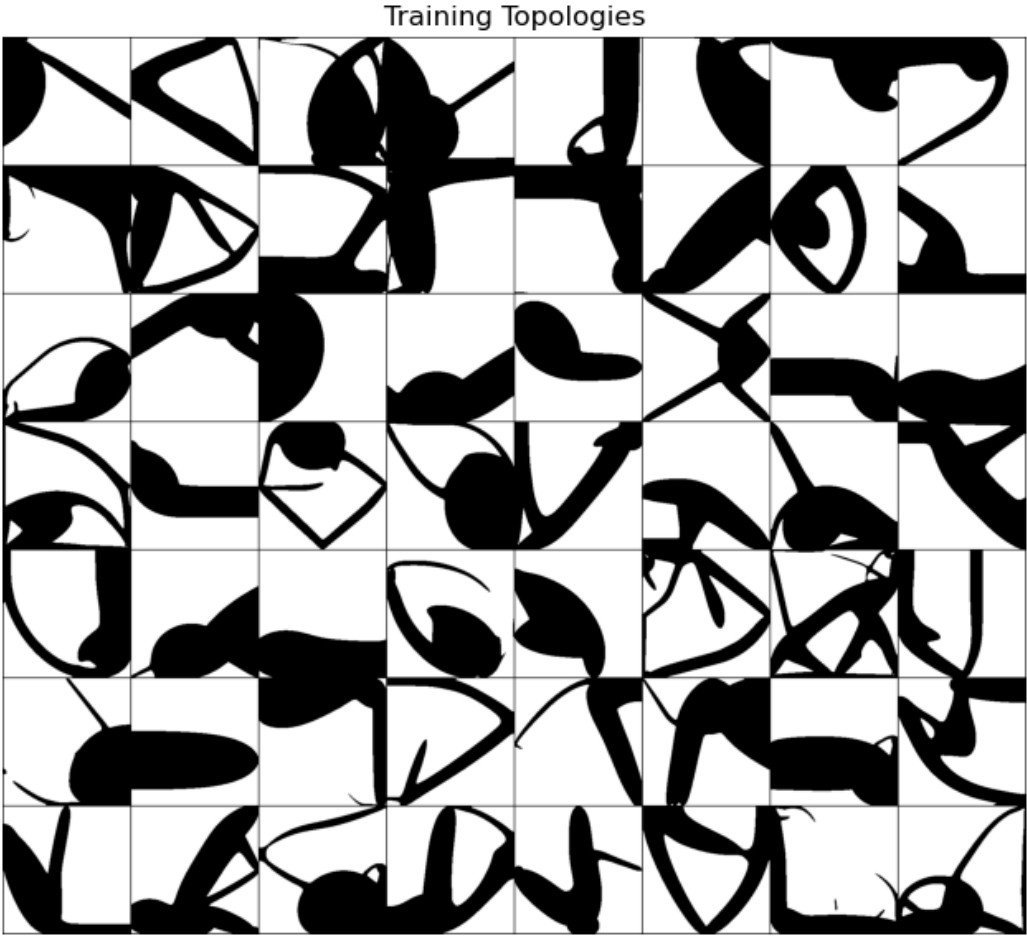

**Figure 17:** Intermediate optimization output after 10 iterations. Resolution: 256x256.

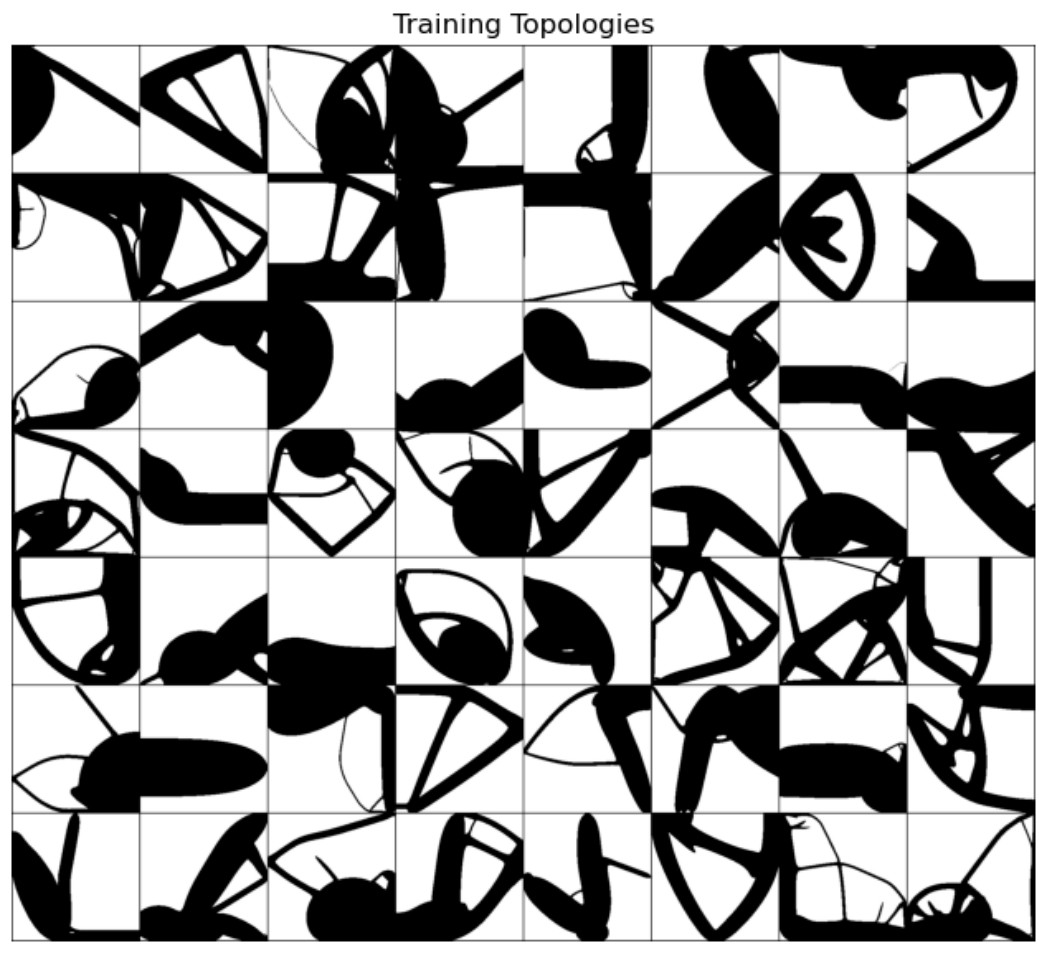

**Figure 18:** Intermediate optimization output after 20 iterations. Resolution: 256x256.

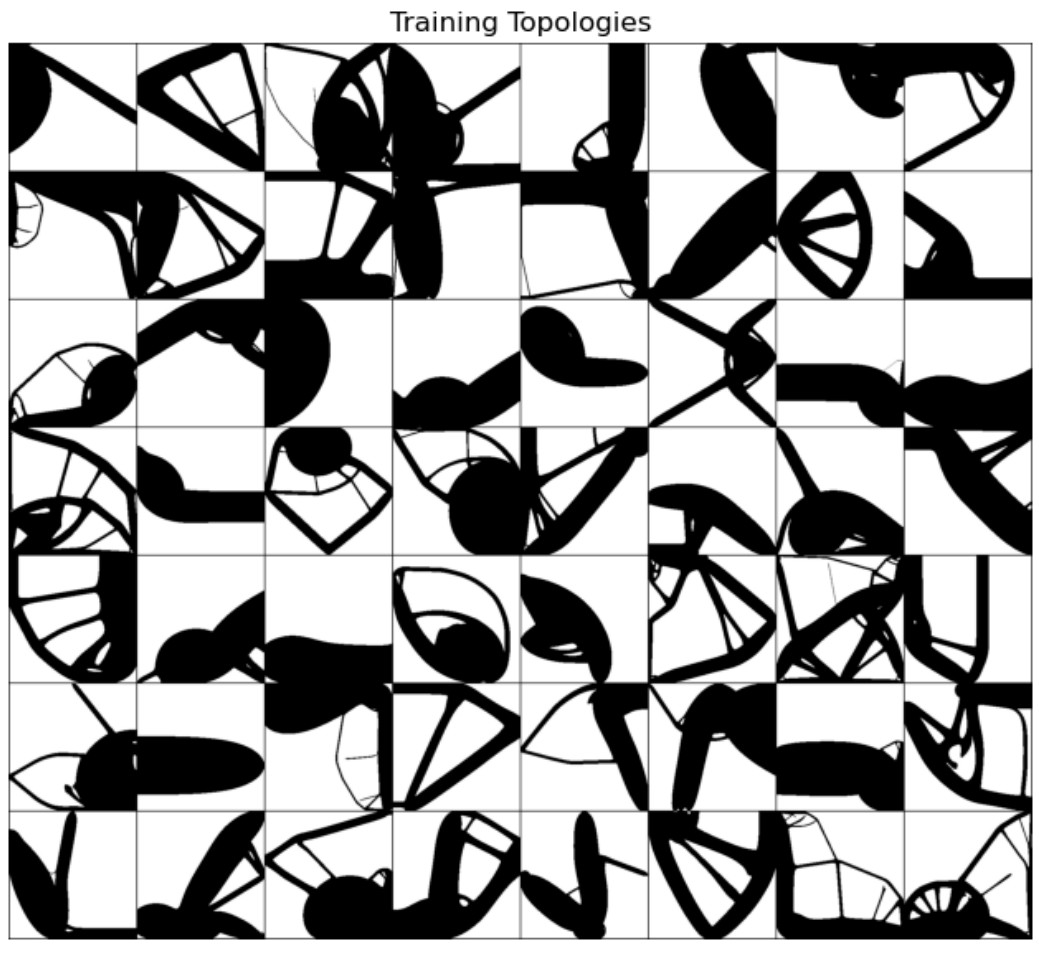

**Figure 19:** Intermediate optimization output after 30 iterations. Resolution: 256x256.

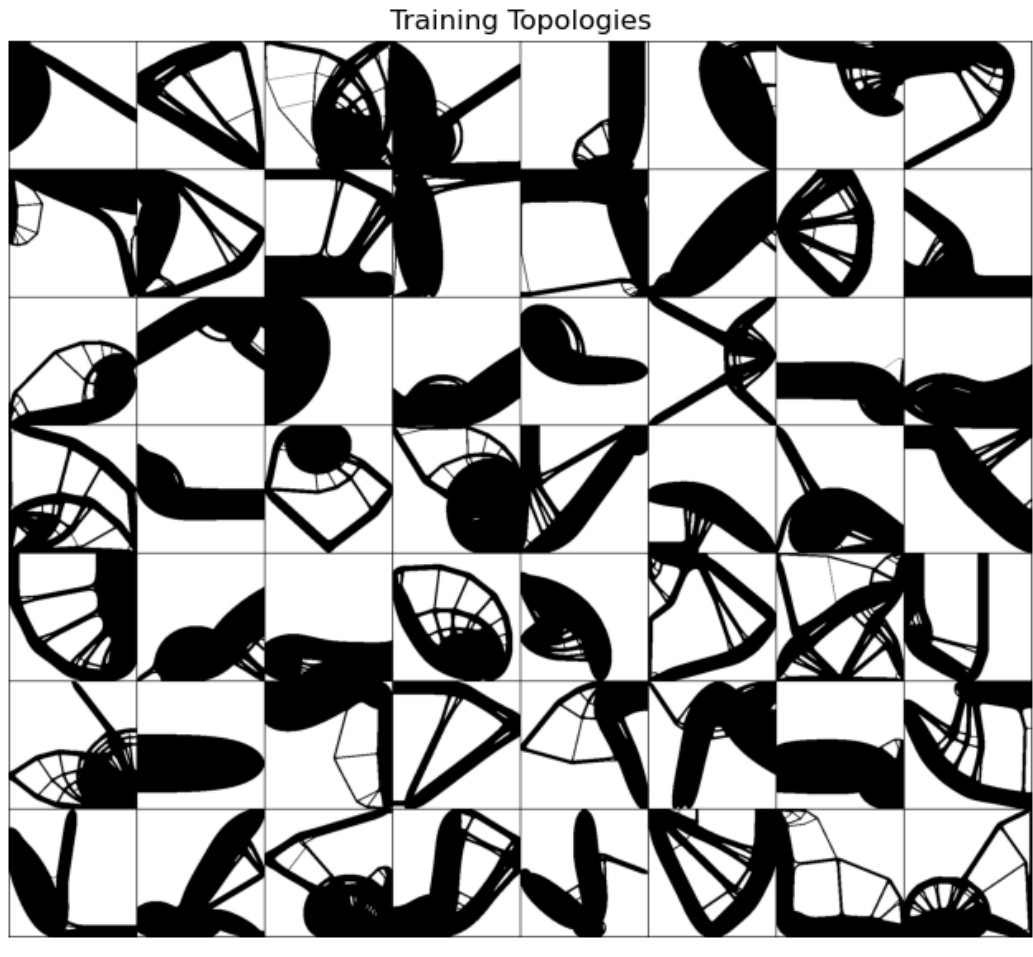

**Figure 20:** Optimized output. Resolution: 256x256.

# G Experimental Details

**Table 11:** Relevant Hyperparameters for baselines and DOM on 64x64 datasets.

|                      | TopoDiff   | TopoDiff-G | DOM w/o TA | DOM w/ TA            |
|----------------------|------------|------------|------------|---------------------|
| Dimension            | 1x64x64    | 1x64x64    | 1x64x64    | 1x64x64             |
| Model Set            | 30k        | 30K        | 30K        | 30K                 |
| Guidance Set         | -          | 150K       | -          | -                   |
| Intermediate Set     | -          | -          | -          | 150K                |
| Test Configurations  | 1800       | 1800       | 1800       | 1800                |
| Batch size           | 64         | 64         | 64         | 64                  |
| Architecture         | Unet       | Unet       | Unet       | Unet                |
| Iterations           | 200K       | 200K       | 200K       | 200K                |
| Learning rate        | $2e^{-4}$  | $2e^{-4}$  | $2e^{-4}$  | $2e^{-4}$           |
| Loss                 | $L_\epsilon$ | $L_\epsilon$ | $L_\epsilon$ | $L_\epsilon + L_{\texttt{TA}}$ |
| Optimizer            | Adam       | Adam       | Adam       | Adam                |

