# OpenReview forum: "Aligning Optimization Trajectories with Diffusion Models for Constrained Design Generation"
_NeurIPS.cc/2023/Conference — NeurIPS 2023 poster_

### Official Review · Reviewer_dj1b · 2023-06-27

**Soundness:** 2 fair
**Presentation:** 2 fair
**Contribution:** 3 good
**Rating:** 6
**Confidence:** 3

**Summary:**

This work studies the use of diffusion models for constrained topology design, a task where physics-based optimization methods still outperform deep learning approaches. The authors introduce diffusion optimization models (DOMs), a diffusion-based approach to generate topology candidates rapidly and efficiently. The main contribution is "trajectory alignment" (TA), which attempts to "align" the intermediate expected predictions (through Tweedie's formula) of the noise/score estimator $\epsilon_\theta(x_t, c)$ with the intermediate topologies of the physics-aware optimization process. The authors evaluate DOM on a dataset of optimized topologies with the SIMP solver, and demonstrate improvements over previous data-driven methods, both in terms of quality and efficiency.

**Edit** Following the rebuttal/discussion period and in light of the changes proposed by the authors, I upgrade my score from 3 to 6.

**Strengths:**

* The subject of generative modeling under constraints is interesting.
* The idea of "aligning" the sampling process of a diffusion model with another process is interesting and worth studying.
* The proposed method demonstrates improvements over previous methods for a dataset of optimized topologies.

**Weaknesses:**

* The proposed method is not mathematically sound. The modified training objective ($\mathcal{L}_{DOM}$) does not lead to a proper diffusion model in the general case, meaning that simulating the reverse process has no guarantees to generate samples from the target distribution. Additionally, aligning trajectories with fundamentally different time axes (optimization steps against perturbation steps) seems dubious and is not formally motivated by the authors.

* I suspect that the proposed method works for the application only because the mapping "constraints -> optimal topology" is (almost) deterministic, leading to a regressive task instead of a generative task. Therefore, the diffusion model learns to transform all noise to the same topology, given the constraints, which explains the very low number of sampling steps. The effectiveness of the method cannot be correctly assessed without a proper generative task. An example of such task would be to generate topologies without conditioning on the constraints, that is sampling from $p(x) = \int p(x, c) ~ dc$ instead of $p(x | c)$. Because the "optimization trajectory" is still available, the proposed method should be applicable. Furthermore, if the application is a generative task, the authors should generate several topologies for each set of constraints (both with DOM and SIMP).

* The link drawn with consistency models (lines 205 and 335) is hazardous. First, consistency models do not "predict their input at each step" as the authors write (line 334) but are forced to be consistent along trajectories, such that they eventually define a mapping "noise -> data". Second, the proposed trajectory alignment pushes all trajectories to be the same, given the constraints, which is vastly different and in no way a "generalization".

* At line 89, the authors claim that DOM "respect constraints". I beleive that neither trajectory alignment nor dense kernel relaxation guarantee the satisfication of constraints. Only the "few-steps direct optimization", which is not part of the diffusion procedure, could guarantee to respect constraints.

* The manuscript has many vague or confusing statements and concepts, which are introduced without definition or context. Follows a non-exhaustive list of such statements: design process, line 4; iterative optimization, line 5; optimization trajectory, line 10; firmly grounded in the underlying physical principles, line 12; two steps, line 13; sample tens or hundreds of layers for each sample, 168; physics-based information, line 192; multi-fidelity mechanisms, line 201; close to the corresponding optimization step in terms of distance, line 204; among those that can be represented by the reverse process, line 217; lines 219-220; transport, line 232; suitable representation, line 232; some mapping, line 234; optimize this loss for the mean values, freeze the mean representations and optimize the variances in a separate step, line 253; volume fraction, line 309; compliance error, line 312. Additionaly, some part of the manuscript are verbose and redundant, notably lines 25-28, the trajectory alignement section, and the lines 234-237.

* There is an overwhelming amount of details related to the application (topology optimization) that distracts the reader from the methodological contribution.

**Questions:**

* At line 315, "(i) In-distribution constraints. The constrainsts in this test set are the same as those of the training set." Does this means that the training and test sets are the same?
* Why are 1-step generation models such as GANs or consistency models slower than 100-steps models?
* When varying the number of sampling steps (e.g. Table 3), are the models trained for that specific number of steps or is there a gap between training and evaluation? If the latter how is that gap handeled?
* What is the proportion of the inference time allocated to pre-processing?
* At line 243, is the $s(t-1)$ formula correct? $s(t) = n_s - \lfloor n_s \frac{t}{T} \rfloor$ seems to match the footnote better.

**Limitations:**

See weaknesses.

---

> ### Author Rebuttal · Authors · 2023-08-08
>
> ## **Authors**: *We would like to clarify that DOM is a well-defined conditional diffusion model. TA is used only during training. At inference time, we sample DOM using standard ancestral sampling and the model does not have access to any intermediate optimization step. See the attached PDF for examples*.
>
> > Strengths:
> The subject of generative modeling under constraints is interesting.
> The idea of "aligning" the sampling process of a diffusion model with another process is interesting and worth studying.
>
> **Authors**: Thanks for mentioning such aspects of our work. The connection between generative models and engineering design is deeply underexplored and we think that trajectory alignment is important for both fields.
>
> >Weaknesses:
> The modified training objective does not lead to a proper diffusion model in the general case, meaning that simulating the reverse process has no guarantees to generate samples from the target distribution.
>
> **Authors**: **DOM is a well-defined conditional diffusion model, trained using a reweighted lower bound to the likelihood**.
>
> * The epsilon parameterization (for each step $t$) is: $w_t || \epsilon_{\theta}(x_t) - \epsilon  ||^2_2$. If $w_t=1$, we reweight the bound giving more importance to large $t$, focusing on global structure (see Alg. 1 in arxiv/2006.11239).
>
> * We train the model variance following IDDPM (Eq.16 arxiv/2102.09672), where at each iteration the model mean is trained using the epsilon-parameterization, then the mean is frozen and the variance is optimized using the ELBO. (lines 247-253)
>
> * The regularization term introduced by TA in Eq.6 is always non-negative and is added to the loss (Eq.7) or equivalently subtracted from the lower bound, providing still a proper objective for DDPM. Notice that we choose the same variance for the distribution over $\tilde x^{\theta}$ and $x^{opt}_{s}$, so TA does not act on the model variance but only on the model mean. Based on these points and the samples, it should be clear that DOM is generative and TA is a regularization mechanism that does not collapse the model to a deterministic mapping.
>
> * **Trajectory alignment is used only during training**. At inference time, we sample DOM using ancestral sampling without any intermediate optimization step. We have undertaken rigorous empirical validation of our model (in-distribution, out-of-distribution, inference efficiency, scalability, generative and engineering metrics).
>
> >The effectiveness of the method cannot be correctly assessed without a proper generative task.
>
> **Authors**: **The provided samples (see PDF and general response) were generated by DOM trained w/ TA and sampled with standard ancestral sampling**. We have also evaluated various generative metrics in Appx B.4 to support our claims. With these samples, we demonstrate that our model generates a diverse range of samples while adhering to the given constraints (10 samples in each row).
>
> >Because the "optimization trajectory" is still available, the proposed method should be applicable.
>
> **Authors**: **This assumption is wrong. During inference, we do not have intermediate optimization steps for the test set**. Our sampling process (at inference time) relies only on conditioning the constraints and the kernels.
> *We understand that the nomenclature ("sampling trajectory") can be confusing and we will be glad to clarify this point in the final version of the paper*.
>
> >The link drawn with consistency models (lines 205 and 335) is hazardous.
>
> **Authors**: Thank you for bringing up this point. **We agree that DOM and CM are different approaches. Our intention is to provide a parallel to build intuition**. Our method, trajectory alignment, does not aim to force all trajectories to be identical. Instead, we strive to bring them as close as possible to each other while learning a generative model for data distribution.
>
> >At line 89, the authors claim that DOM "respect constraints".
>
> **Authors**: **DOM w/ TA is a robust and effective model for topology optimization**. While numerical solvers provide guarantees, through empirical evaluation our model consistently generates designs that largely adhere to the imposed constraints while achieving high levels of performance.
>
> > Questions: Does this means that the training and test sets are the same?
>
> **Authors**: **Training and test conditioning configurations are different**. The in-distro scenario poses a challenging generalization task because the TO problem is highly non-linear. SOTA GANs generate >50% of samples that do not fulfill the constraints in the in-distro scenario. In the in-distro test set, boundary conditions match those from the training set, but loadings and volume fractions differ. This is standard evaluation practice for generative models (as proposed in TopoDiff).
>
>
> > Why are 1-step generation models such as GANs or consistency models slower than 100-steps models?
>
> **Authors**: **GANs are slower because of the field computations during pre-processing. Without pre-processing, GANs perform extremely poorly for topology optimization**.
> In Table 2, we sample consistency models with 100 steps to have a fair evaluation. We also show that CM underperforms compared to DOM in appx Table 2.
>
> > When varying the number of sampling steps (e.g. Table 3), are the models trained for that specific number of steps or is there a gap between training and evaluation? If the latter how is that gap handeled?
>
> **Authors**: **No, we train one model with T=1000 and then vary the sampling steps at sampling time**. We use the IDDPM skipping approach (w/o using DDIM).
>
> > What is the proportion of the inference time allocated to pre-processing?
>
> **Authors**: **For TopoDiff, >50% for 64x64 res, and >70% for 256x256 res. For DOM is <1%**. In appx Table 3 we show this proportion, showing the advantage of our method. Pre-processing is time-consuming and problem-specific, whereas our approach is cheap and applicable to a general set of vision-based problems.

---

> > ### Comment · Reviewer_dj1b · 2023-08-10
> >
> > Thank you for your answers. Some of my concerns were addressed, but there seems to be a misunderstanding on the major one(s). I already know that TA is only used during training. My issue is that the objective $L_{DOM}$ (Eq. 6 and 7) does not have the same optimum as the DDPM objective $L_{t-1, \epsilon}$ due to the regularization term $L^{TA}$. Therefore, minimizing this objective cannot lead to a proper diffusion model, that is a model of the score $\nabla_{x_t} \log p(x_t | c)$. It is possible that minimizing $\mathcal{L}_{DOM}$ leads to another (non-diffusion) stochastic process between $\mathcal{N}(0, I)$ and $p(x_0 | c)$, but without theoretical evidence, it is hard to believe.
> >
> > My intuition is that TA(-CLEAN) works in this case because $p(x_0 | c)$ is a degenerate (low variance) distribution. Otherwise, forcing all sampling trajectories (given constraints $c$) to be, citing your rebuttal, "as close as possible to each other" would result in some issues. In fact, we can see in Appendix B.2 that only TA-CLEAN leads to improvements over the NO-TA variant, while other TA modes (MULTI, NOISY, PERF) lead to degradation of compliance, which would not be the case is the approach was sound.
> >
> > This is why I asked you in my review to train/apply DOM w/ TA without conditioning on the constraints, that is training solely from topologies $x_0 \sim p(x_0) = E_{p(c)} [p(x_0 | c)]$ and their optimization trajectory, which would undoubtedly be a proper generation task.
> >
> > **However**, the additional examples of generation that you provide seem to indicate that DOM w/ TA generates different topologies given the same constraints. So I am willing to give the benefit of the doubt here. If you to add in the manuscript that you **don't provide theoretical evidence** for the soundness of TA, but demonstrate **empirically** that TA works for your task, I would agree to raise my score. Nevertheless, I would prefer theoretical developments.
> >
> > Additionally, like reviewer hnNz, I mentionned in my review that some parts (notably Section 3) of the manuscript are hard to follow, especially as it introduces concepts without definitions. I would strongly advise clarifying these parts and further dissociating technical contributions from domain contributions.
> >
> > Finally, I have some issues with the Appendix B.4, which you mentioned in the rebuttal. I don't see how the IS and FID metrics are relevant for you task. They are only relevant for "natural" images (RBG) as they rely on the Inception network trained on ImageNet. Also, for FID, what distributions did you compare? $p(x_0 | c)$ and $p_\theta(x_0 | c)$ ? $p(x_0) = E_{p(c)}[p(x_0 | c)]$ and $p_\theta(x_0) = E_{p(c)}[p_\theta(x_0 | c)]$ ? I could not find relevant code in the supplementary material.

---

> > > ### Author Response · Authors · 2023-08-10
> > > **Response Reviewer dj1b**
> > >
> > > ### **Authors**: *Thank you for thoroughly reviewing the examples and the sampling process. We appreciate your suggestion: our contribution is indeed empirical and at the intersection between engineering design and generative models*.
> > >
> > > ### **We would be happy to clearly mention in the manuscript, that the focus of this work is not theoretical evidence for TA, but demonstrate empirically that TA works for TO and its vast impact on problems in engineering design with hard constraints and performance requirements**.
> > >
> > > -----------
> > >
> > > >Thank you for your answers. Some of my concerns were addressed, but there seems to be a misunderstanding on the major one(s). I already know that TA is only used during training.
> > >
> > > **Authors**: We appreciate your prompt response and your open-mindedness toward our work and contributions. We are thankful for the opportunity to further elucidate our approach following the rebuttal.
> > >
> > >
> > > >However, the additional examples of generation that you provide seem to indicate that DOM w/ TA generates different topologies given the same constraints. If you to add in the manuscript that you don't provide theoretical evidence for the soundness of TA, but demonstrate empirically that TA works for your task, I would agree to raise my score.
> > >
> > > **Authors:** We appreciate your suggestion: our contribution is indeed empirical and at the intersection between engineering design and generative models.
> > > **We are happy to clearly state in our paper that we do not provide theoretical evidence for TA, but demonstrate empirically that TA works**. It is true that in this work, our focus is on demonstrating the empirical effectiveness of TA. We hope that our strong empirical results will open a new line of research at the intersection between engineering design and generative models.
> > >
> > > > My issue is that the objective
> > >  (Eq. 6 and 7) does not have the same optimum as the DDPM objective
> > >  due to the regularization term
> > >
> > > **Authors**: Where we agree that our evidence for TA is empirical, we would like to provide some intuition regarding why it works. Consider the case where we have the same number of steps in the sampling trajectory and the optimization trajectory $s(t) = t$. In this setting, we should be able to match each step exactly. At each step, we can search for $\tilde x^{\theta}$. If the model $\epsilon_{\theta}$ can represent the optimization trajectory, then the regularization term can be fully satisfied. After training, the sampling and optimization trajectories should be close.
> > >
> > > >I would strongly advise clarifying these parts and further dissociating technical contributions from domain contributions.
> > >
> > > **Authors:** Thank you for your suggestion. **We agree with the reviewers and we will better dissociate technical contributions from domain contributions**. As we operate at the intersection of engineering design and generative models, we encountered the complexity of balancing comprehensive background information with conciseness, and we are committed to better separating technical background and methodological contribution. We hope that our contribution will outweigh these aspects, providing valuable insights and methods to the community.
> > >
> > > >Finally, I have some issues with the Appendix B.4, which you mentioned in the rebuttal. I don't see how the IS and FID metrics are relevant for you task.
> > >
> > > **Authors:** Thank you for pointing this out. We fully agree with you! **We compute such metrics to provide evidence that the standard way to evaluate diffusion models (FID, IS, P, R) is not appropriate for generative engineering design**. From our appx (lines 87-91):
> > >
> > > * *it is challenging to gain any understanding just by relying on classic generative metrics when evaluating
> > > constrained design generation. These results justify the need for an evaluation that considers the performance and feasibility of the generated design*.
> > >
> > > The primary motivation behind computing these metrics is to underscore their lack of reliability and utility for engineering design. We calculate these metrics both for the sake of thoroughness and to perform ablation analysis. It's worth noting that even within engineering design, conventional generative models for TO have been assessed using simple metrics such as pixel-wise distance. This practice has disregarded critical design aspects like **constraint fulfillment, performance criteria, manufacturability, and feasibility, all factors that are at the core of our evaluation**.
> > >
> > > >Also, for FID, what distributions did you compare?
> > >
> > > **Authors:** We compare a held-out set from $p(x_0)$ obtained using SIMP with $p_{\theta}(x_0)$. In the dataset of 2d TO that we are releasing with the paper, we also have 100K topologies w/o constraints. We select 10K topologies from this set and use them to compute (FID, IS, P, R) for all the baselines and DOM. We sample 10K samples using 100 steps from each model. We use the code provided in `github/openai/guided-diffusion/tree/main/evaluations`.

---

> > > > ### Comment · Reviewer_dj1b · 2023-08-11
> > > >
> > > > > If the model $\epsilon_\theta$ can represent the optimization trajectory, then the regularization term can be fully satisfied.
> > > >
> > > > The issue is that it is not possible to satisfy both the denoising diffusion term and the regularization term **at the same time**.
> > > >
> > > > Anyway, as you agree with my suggestion to state that this work does not provide theoretical evidence for TA, but demonstrate empirically that TA works for TO, I agree to raise my score from 3 to 5. Theoretical evidence and/or experiments applying DOM w/ TA without conditioning on the constraints can convince me to raise my score higher.

---

> > > > > ### Author Response · Authors · 2023-08-13
> > > > > **Additional Experiment for DOM w/ TA w/ and w/o Conditioning**
> > > > >
> > > > > ### **Authors**:  *We provide experiments on conditional and unconditional samples below. We provide distance matrices between samples because we cannot upload images in the comments*.
> > > > >
> > > > > -----------
> > > > >
> > > > > >I agree to raise my score from 3 to 5
> > > > >
> > > > > **Authors**: Thank you for appreciating our contribution.
> > > > >
> > > > > >experiments applying DOM w/ TA without conditioning on the constraints can convince me to raise my score higher.
> > > > >
> > > > > **Authors**: **We are happy to provide additional experiments for DOM trained with TA and remove the conditioning on the constraints**. Two important points:
> > > > >
> > > > > * **Volume Fraction**. We remove the conditioning on boundary conditions, loading, and kernels, setting the corresponding conditioning channels to zero (the DOM architecture is an Unet).
> > > > > However, for the conditioning on volume fraction (the % of the starting domain that should be used for the generative design), there is no easy way to remove it: if we set the volume fraction to zero, the model will try to remove as much material as possible and, in absence of constraints, we will obtain a final design with very little material. For this reason, we set VF to fix values (0, 0.5) or random.
> > > > > **We remain open to conducting further experiments and exploring additional metrics to enhance our approach**.
> > > > >
> > > > > * **We cannot upload images or share links to samples in this phase**. Here we can report some metrics related to the `diversity of the generated design in the absence of constraints`: in particular, we report the (L2) distance matrices between 10 different samples. d(i, j) is the distance between sample_i in row i and sample_j in col j. **We will include unconditional samples in the appendix**.
> > > > >
> > > > > ---------------------
> > > > > ## Experiments
> > > > >
> > > > > 1. In the first **table (1)**, we report the distance between 10 **conditional samples** (like the ones that we provide in the PDF).
> > > > > 2. In the second **table (2)**, we report the distance between 10 **unconditional samples**, fixing VF=0.5 and **removing the boundary, loading, and kernel constraints**.
> > > > > 3. In the third **table (3)**, we report the distance between 10 **unconditional samples**, fixing VF=0.0 and **removing the boundary, loading, and kernel constraints**.
> > > > > 4. In the fourth **table (4)**, we report the distance between 10 **unconditional samples**, fixing VF=0.5 and using random numbers from a N(0, 1) as conditioning for boundary, loading, and kernel constraints.
> > > > > 5. In the fifth **table (5)**, we report the distance between 10 **unconditional samples**, using random numbers from a N(0, 1) as conditioning for boundary, loading, and kernel, and volume fraction constraints.
> > > > >
> > > > > ------------------
> > > > >
> > > > > ### **Table (1)** - Pixel-wise Distance matrix between 10 **conditional samples** (first row in the attached PDF). All generated pixel values in [0, 1]. Domain 64x64.
> > > > >
> > > > > The distance between conditional samples (DOM w/ TA w/ constraints) changes for all the considered 10 samples. All 10 samples are conditioned with the same constraints. In particular, the empirical mean and standard deviation for each row (removing the sample itself) are different. **This table provides evidence for variability in the conditional samples, as illustrated in the attached pdf**. cs_i: conditional sample i. It also corroborates the fact that DOM is a generative model.
> > > > >
> > > > > * Generated samples description: see attached pdf in the general rebuttal.
> > > > >
> > > > > |     |   cs_0 |   cs_1 |   cs_2 |   cs_3 |   cs_4 |   cs_5 |   cs_6 |   cs_7 |   cs_8 |   cs_9 |   MEAN |   SD |
> > > > > |-----|-------|-------|-------|-------|-------|-------|-------|-------|-------|-------|--------|------|
> > > > > | **cs_0**  |  0.00 | 16.34 | 12.25 | 14.83 | 15.68 | 17.32 | 14.56 | 11.92 | 17.58 | 12.77 |  14.81 | 2.01 |
> > > > > | **cs_1**  | 16.34 |  0.00 | 14.53 | 11.27 | 13.15 | 13.89 | 13.38 | 13.53 | 11.92 | 12.96 |  13.44 | 1.38 |
> > > > > | **cs_2**  | 12.25 | 14.53 |  0.00 | 14.63 | 13.71 | 13.27 | 10.20 | 10.58 | 15.26 |  9.33 |  12.64 | 2.03 |
> > > > > | **cs_3**  | 14.83 | 11.27 | 14.63 |  0.00 | 13.93 | 14.21 | 13.56 | 13.49 | 12.45 | 12.77 |  13.46 | 1.07 |
> > > > > | **cs_4**  | 15.68 | 13.15 | 13.71 | 13.93 |  0.00 | 14.14 | 12.41 | 12.08 | 13.00 | 12.21 |  13.37 | 1.08 |
> > > > > | **cs_5**  | 17.32 | 13.89 | 13.27 | 14.21 | 14.14 |  0.00 | 10.58 | 13.19 | 11.27 | 13.60 |  13.50 | 1.80 |
> > > > > | **cs_6**  | 14.56 | 13.38 | 10.20 | 13.56 | 12.41 | 10.58 |  0.00 | 11.49 | 13.38 | 10.34 |  12.21 | 1.52 |
> > > > > | **cs_7**  | 11.92 | 13.53 | 10.58 | 13.49 | 12.08 | 13.19 | 11.49 |  0.00 | 13.60 |  8.54 |  12.05 | 1.59 |
> > > > > | **cs_8**  | 17.58 | 11.92 | 15.26 | 12.45 | 13.00 | 11.27 | 13.38 | 13.60 |  0.00 | 13.64 |  13.57 | 1.78 |
> > > > > | **cs_9**  | 12.77 | 12.96 |  9.33 | 12.77 | 12.21 | 13.60 | 10.34 |  8.54 | 13.64 |  0.00 |  11.80 | 1.79 |

---

> > > > > > ### Author Response · Authors · 2023-08-13
> > > > > > **Table (2) - distance between unconditional samples - fixing VF=0.5 and removing the boundary, loading, and kernel constraints.**
> > > > > >
> > > > > > ### **Table (2)** - Pixel-wise Distance matrix between 10 **unconditional samples** fixing VF=0.5 and removing the boundary, loading, and kernel constraints. All generated pixel values in [0, 1]. Domain 64x64.
> > > > > >
> > > > > > The distance between unconditional samples (DOM w/ TA w/o constraints) changes for all the considered samples. **The distance between unconditional samples is much larger than the one between conditional samples in Table (1), as expected. At the same time, there is variability in such distances, corroborating our argument that DOM is a proper generative model**. s_i: unconditional sample i. DOM without constraints generates diverse samples.
> > > > > >
> > > > > > * Generated samples description: given that we do not have constraints and we require VF=0.5, DOM generates diverse shapes that in some cases are not connected with the boundaries.
> > > > > >
> > > > > > |     |   s_0 |   s_1 |   s_2 |   s_3 |   s_4 |   s_5 |   s_6 |   s_7 |   s_8 |   s_9 |   MEAN |   SD |
> > > > > > |-----|-------|-------|-------|-------|-------|-------|-------|-------|-------|-------|--------|------|
> > > > > > | **s_0** |  0.00 | 29.46 | 36.19 | 37.99 | 41.90 | 24.78 | 37.40 | 32.30 | 33.39 | 24.06 |  33.05 | 5.72 |
> > > > > > | **s_1** | 29.46 |  0.00 | 43.08 | 34.83 | 50.73 | 36.88 | 42.41 | 15.33 | 35.31 | 35.17 |  35.91 | 9.32 |
> > > > > > | **s_2** | 36.19 | 43.08 |  0.00 | 43.69 | 31.05 | 37.47 | 40.16 | 45.27 | 31.89 | 31.22 |  37.78 | 5.27 |
> > > > > > | **s_3** | 37.99 | 34.83 | 43.69 |  0.00 | 51.25 | 39.94 | 35.97 | 35.07 | 40.27 | 42.80 |  40.20 | 4.93 |
> > > > > > | **s_4** | 41.90 | 50.73 | 31.05 | 51.25 |  0.00 | 39.55 | 43.19 | 52.72 | 40.21 | 36.78 |  43.04 | 6.87 |
> > > > > > | **s_5** | 24.78 | 36.88 | 37.47 | 39.94 | 39.55 |  0.00 | 31.03 | 37.93 | 40.71 | 30.18 |  35.39 | 5.14 |
> > > > > > | **s_6** | 37.40 | 42.41 | 40.16 | 35.97 | 43.19 | 31.03 |  0.00 | 41.98 | 44.07 | 41.98 |  39.80 | 3.99 |
> > > > > > | **s_7** | 32.30 | 15.33 | 45.27 | 35.07 | 52.72 | 37.93 | 41.98 |  0.00 | 37.47 | 38.18 |  37.36 | 9.64 |
> > > > > > | **s_8** | 33.39 | 35.31 | 31.89 | 40.27 | 40.21 | 40.71 | 44.07 | 37.47 |  0.00 | 32.43 |  37.31 | 4.04 |
> > > > > > | **s_9** | 24.06 | 35.17 | 31.22 | 42.80 | 36.78 | 30.18 | 41.98 | 38.18 | 32.43 |  0.00 |  34.76 | 5.64 |

---

> > > > > > > ### Author Response · Authors · 2023-08-13
> > > > > > > **Table (3) - distance between unconditional samples - fixing VF=0.0 and removing the boundary, loading, and kernel constraints.**
> > > > > > >
> > > > > > > ### **Table (3)** - Pixel-wise Distance matrix between 10 unconditional samples fixing VF=0.0 and removing the boundary, loading, and kernel constraints. All generated pixel values in [0, 1]. Domain 64x64.
> > > > > > >
> > > > > > > The distance between unconditional samples (DOM w/ TA w/o constraints) changes for all the considered samples.
> > > > > > > This setting with VF=0.0 is a degenerate case: DOM will try to remove as much material as possible given the volume fraction requirement and the absence of constraints. **You can see that the distances between samples are much smaller than the ones in Table (2). This is an artifact because most of the pixels (~97.5 %) are zeros (absence of material)**.
> > > > > > > At the same time, there is variability in such distances, corroborating our argument that DOM is a proper generative model. s_i: unconditional sample i. DOM without constraints generates diverse samples.
> > > > > > >
> > > > > > > * Generated samples description: given that we do not have constraints and we require VF=0.0, DOM generates diverse curves with most of the domain empty.
> > > > > > >
> > > > > > > |     |   s_0 |   s_1 |   s_2 |   s_3 |   s_4 |   s_5 |   s_6 |   s_7 |   s_8 |   s_9 |   MEAN |   SD |
> > > > > > > |-----|-------|-------|-------|-------|-------|-------|-------|-------|-------|-------|--------|------|
> > > > > > > | **s_0** |  0.00 | 11.79 | 14.59 | 15.87 | 14.42 | 12.61 | 14.35 | 13.82 | 13.71 | 14.32 |  13.94 | 1.11 |
> > > > > > > | **s_1** | 11.79 |  0.00 | 13.19 | 13.96 | 13.38 | 11.49 | 12.69 | 12.00 | 12.21 | 12.81 |  12.61 | 0.77 |
> > > > > > > | **s_2** | 14.59 | 13.19 |  0.00 | 15.72 | 14.87 | 14.42 | 12.61 | 12.25 | 14.93 | 14.97 |  14.17 | 1.13 |
> > > > > > > | **s_3** | 15.87 | 13.96 | 15.72 |  0.00 | 16.97 | 15.59 | 15.56 | 14.73 | 14.83 | 16.46 |  15.52 | 0.86 |
> > > > > > > | **s_4** | 14.42 | 13.38 | 14.87 | 16.97 |  0.00 | 14.18 | 14.70 | 14.11 | 14.97 | 13.67 |  14.58 | 0.98 |
> > > > > > > | **s_5** | 12.61 | 11.49 | 14.42 | 15.59 | 14.18 |  0.00 | 14.18 | 13.71 | 13.23 | 13.34 |  13.64 | 1.10 |
> > > > > > > | **s_6** | 14.35 | 12.69 | 12.61 | 15.56 | 14.70 | 14.18 |  0.00 | 12.12 | 14.70 | 14.66 |  13.95 | 1.11 |
> > > > > > > | **s_7** | 13.82 | 12.00 | 12.25 | 14.73 | 14.11 | 13.71 | 12.12 |  0.00 | 14.25 | 14.00 |  13.44 | 0.97 |
> > > > > > > | **s_8** | 13.71 | 12.21 | 14.93 | 14.83 | 14.97 | 13.23 | 14.70 | 14.25 |  0.00 | 14.73 |  14.17 | 0.89 |
> > > > > > > | **s_9** | 14.32 | 12.81 | 14.97 | 16.46 | 13.67 | 13.34 | 14.66 | 14.00 | 14.73 |  0.00 |  14.33 | 1.00 |

---

> > > > > > > > ### Author Response · Authors · 2023-08-13
> > > > > > > > **Table (4) - distance between unconditional samples - fixing VF=0.5 and using random values from a N(0, 1) for boundary, loading, and kernel constraints.**
> > > > > > > >
> > > > > > > > ### **Table (4)** - Pixel-wise Distance matrix between 10 unconditional samples fixing VF=0.5 and using random values from a N(0, 1) for the boundary, loading, and kernel constraints. All generated pixel values in [0, 1]. Domain 64x64.
> > > > > > > >
> > > > > > > >
> > > > > > > > The distance between unconditional samples (DOM w/ TA w/o constraints) changes for all the considered samples. **The distance between unconditional samples is much larger than the one between conditional samples in Table (1), as expected. Notice that distances are even higher than in Table (2), showing that the model is having issues dealing with the noise. At the same time, there is variability in such distances, corroborating our argument that DOM is a proper generative model**. s_i: unconditional sample i. DOM without constraints generates diverse samples.
> > > > > > > >
> > > > > > > > * Generated samples description: given that we require VF=0.5 and the model is challenged with noisy and uninformative conditioning, DOM generates diverse samples with no clear structure, floating material, and in general unfeasible.
> > > > > > > >
> > > > > > > > |     |   s_0 |   s_1 |   s_2 |   s_3 |   s_4 |   s_5 |   s_6 |   s_7 |   s_8 |   s_9 |   MEAN |    SD |
> > > > > > > > |-----|-------|-------|-------|-------|-------|-------|-------|-------|-------|-------|--------|-------|
> > > > > > > > | **s_0** |  0.00 | 53.54 | 40.99 | 55.09 | 39.65 | 46.69 | 36.61 | 55.55 | 48.67 | 46.48 |  47.03 |  6.52 |
> > > > > > > > | **s_1** | 53.54 |  0.00 | 46.29 | 25.57 | 53.09 | 39.96 | 50.19 | 28.86 | 37.36 | 40.06 |  41.66 |  9.50 |
> > > > > > > > | **s_2** | 40.99 | 46.29 |  0.00 | 41.89 | 49.01 | 36.58 | 36.77 | 41.33 | 35.79 | 32.65 |  40.14 |  4.95 |
> > > > > > > > | **s_3** | 55.09 | 25.57 | 41.89 |  0.00 | 53.86 | 36.32 | 46.49 | 18.73 | 29.33 | 36.26 |  38.17 | 11.72 |
> > > > > > > > | **s_4** | 39.65 | 53.09 | 49.01 | 53.86 |  0.00 | 47.05 | 36.69 | 53.61 | 53.84 | 53.10 |  48.88 |  6.19 |
> > > > > > > > | **s_5** | 46.69 | 39.96 | 36.58 | 36.32 | 47.05 |  0.00 | 36.80 | 35.13 | 31.67 | 30.33 |  37.84 |  5.52 |
> > > > > > > > | **s_6** | 36.61 | 50.19 | 36.77 | 46.49 | 36.69 | 36.80 |  0.00 | 46.30 | 43.49 | 42.52 |  41.76 |  4.94 |
> > > > > > > > | **s_7** | 55.55 | 28.86 | 41.33 | 18.73 | 53.61 | 35.13 | 46.30 |  0.00 | 28.72 | 35.97 |  38.25 | 11.47 |
> > > > > > > > | **s_8** | 48.67 | 37.36 | 35.79 | 29.33 | 53.84 | 31.67 | 43.49 | 28.72 |  0.00 | 26.63 |  37.28 |  8.98 |
> > > > > > > > | **s_9** | 46.48 | 40.06 | 32.65 | 36.26 | 53.10 | 30.33 | 42.52 | 35.97 | 26.63 |  0.00 |  38.22 |  7.80 |

---

> > > > > > > > > ### Author Response · Authors · 2023-08-13
> > > > > > > > > **Table (5) - distance between unconditional samples - using random values from a N(0, 1) for boundary, loading, kernel, and volume fraction constraints.**
> > > > > > > > >
> > > > > > > > > ### **Table (5)** - Pixel-wise Distance matrix between 10 unconditional samples using random values from a N(0, 1) for the boundary, loading, and kernel, and volume fraction constraints. All generated pixel values in [0, 1]. Domain 64x64.
> > > > > > > > >
> > > > > > > > > The distance between unconditional samples (DOM w/ TA w/o constraints) changes for all the considered samples. **The distance between unconditional samples is much larger than the one between conditional samples in Table (1), as expected. At the same time, there is variability in such distances, corroborating our argument that DOM is a proper generative model**. s_i: unconditional sample i. DOM without constraints generates diverse samples.
> > > > > > > > >
> > > > > > > > > * Generated samples description: the model is challenged with noisy and uninformative conditioning, but it should have more "flexibility" because we do not require a fixed VF. DOM generates diverse samples with no clear structure, floating material, and in general unfeasible.
> > > > > > > > >
> > > > > > > > > |     |   s_0 |   s_1 |   s_2 |   s_3 |   s_4 |   s_5 |   s_6 |   s_7 |   s_8 |   s_9 |   MEAN |   SD |
> > > > > > > > > |-----|-------|-------|-------|-------|-------|-------|-------|-------|-------|-------|--------|------|
> > > > > > > > > | **s_0** |  0.00 | 25.92 | 42.87 | 42.52 | 43.76 | 34.61 | 45.37 | 38.76 | 35.24 | 33.90 |  38.11 | 5.91 |
> > > > > > > > > | **s_1** | 25.92 |  0.00 | 38.42 | 42.80 | 42.30 | 38.78 | 47.31 | 40.89 | 40.30 | 35.93 |  39.18 | 5.58 |
> > > > > > > > > | **s_2** | 42.87 | 38.42 |  0.00 | 45.76 | 45.27 | 37.09 | 47.69 | 32.16 | 46.43 | 37.91 |  41.51 | 5.01 |
> > > > > > > > > | **s_3** | 42.52 | 42.80 | 45.76 |  0.00 | 39.41 | 41.64 | 40.47 | 45.76 | 41.38 | 45.20 |  42.77 | 2.20 |
> > > > > > > > > | **s_4** | 43.76 | 42.30 | 45.27 | 39.41 |  0.00 | 43.05 | 39.03 | 43.35 | 34.54 | 44.29 |  41.67 | 3.20 |
> > > > > > > > > | **s_5** | 34.61 | 38.78 | 37.09 | 41.64 | 43.05 |  0.00 | 44.90 | 37.42 | 39.72 | 38.28 |  39.50 | 3.03 |
> > > > > > > > > | **s_6** | 45.37 | 47.31 | 47.69 | 40.47 | 39.03 | 44.90 |  0.00 | 43.20 | 40.45 | 45.75 |  43.79 | 2.99 |
> > > > > > > > > | **s_7** | 38.76 | 40.89 | 32.16 | 45.76 | 43.35 | 37.42 | 43.20 |  0.00 | 38.81 | 34.86 |  39.47 | 4.09 |
> > > > > > > > > | **s_8** | 35.24 | 40.30 | 46.43 | 41.38 | 34.54 | 39.72 | 40.45 | 38.81 |  0.00 | 35.31 |  39.13 | 3.54 |
> > > > > > > > > | **s_9** | 33.90 | 35.93 | 37.91 | 45.20 | 44.29 | 38.28 | 45.75 | 34.86 | 35.31 |  0.00 |  39.05 | 4.47 |

---

> > > > > > > > > > ### Comment · Reviewer_dj1b · 2023-08-15
> > > > > > > > > >
> > > > > > > > > > Thank you for these additional experiments. They indeed strengthen the claim that DOM /w TA is a generative model. I will update my score to 6. Note however that generating diverse samples does not mean that DOM is a proper diffusion model of $p(x_0 | c)$ (it could be another, relatively similar distribution). So it is still necessary to mention that your work does not provide theoretical evidence for TA.

---

> > > > > > > > > > > ### Author Response · Authors · 2023-08-15
> > > > > > > > > > > **Remarks to Reviewer dj1b**
> > > > > > > > > > >
> > > > > > > > > > > > They indeed strengthen the claim that DOM /w TA is a generative model. I will update my score to 6.
> > > > > > > > > > >
> > > > > > > > > > > **Authors**: Thank you. We appreciated the fast response and the possibility to clarify our contribution with the rebuttal and additional experiments.
> > > > > > > > > > >
> > > > > > > > > > > > So it is still necessary to mention that your work does not provide theoretical evidence for TA.
> > > > > > > > > > >
> > > > > > > > > > > **Authors**: Yes, we totally agree. We appreciated your feedback and we commit to:
> > > > > > > > > > >
> > > > > > > > > > > * Clearly state in the manuscript, that **the focus of this work is not theoretical evidence for TA, but demonstrate empirically that TA works for TO and its vast impact on problems in engineering design with hard constraints and performance requirements**. We will also add more intuition about why TA works.
> > > > > > > > > > >
> > > > > > > > > > > * Better separate technical contributions from domain contributions.
> > > > > > > > > > >
> > > > > > > > > > > * Add visualizations of conditional (attached pdf) and unconditional samples (for the additional experiments) in the appendix.

---

### Official Review · Reviewer_hnNz · 2023-06-28

**Soundness:** 3 good
**Presentation:** 2 fair
**Contribution:** 2 fair
**Rating:** 5
**Confidence:** 3

**Summary:**

The paper presents a diffusion-based method for topology optimization. The method seems to base on TopoDiff, but uses trajectory alignment instead of guidance to produce plausible results even under few-step sampling.

**Strengths:**

Diffusion models seem like a natural fit for solving inverse problems, but it's still open what is the most efficient and practical way to do so. This paper explores trajectory alignment as an alternative to guidance.

**Weaknesses:**

The paper doesn't read very well, especially section 3. It's clearly meant as an extension to TopoDiff, and it's often hard to figure out which aspect of it is novel, and what is previous work. Figure 1, 2 don't really help for understanding the method. The description of kernel relaxation (both in the main text and appendix) is very hard to understand without any background in topology optimization. I assume the 'simulation trajectory' comes from running the simulator S (which I assume is SIMP?), but this doesn't seem to be described anywhere. In general, the paper assumes a lot of domain knowledge, and doesn't do enough to explain its relevance to the ML community. While there's plenty of room to study the intersection of engineering problems and ML at NeurIPS, the key interest of the community is ML methods which can be applied widely-- this is a bit of a missed opportunity. Finally, related work absolutely needs to be part of the main text and can't be moved into the appendix.

I'm not a domain expert in topology optimization, and It's a bit hard for me to judge the relevance of the results. The main point of comparison seems to be TopoDiff; visually, the results in the appendix seem very similar, and scores in Tab. 2 are very similar as well. OOD generalization without SIMP seems even worse than TopoDiff. The post-processing with a few steps of SIMP seems a bit sketchy-- you could presumably do this with the output of other models as well, and the whole point of the paper is to avoid running SIMP. Relatedly, it would be great to actually compare performance against a GT solver like SIMP, as the method the paper wants to replace.
The main benefit of DOM seems to be inference speed as it can produce reasonable designs after very few diffusion steps. If this is the case though, the paper needs to make a more concerted argument and comparisons about inference time, and why that matters. It would e.g. be great to see a line plot with x axis "wall clock inference time" and y axis "score" for both DOM and Topodiff. Fig 6 is interesting, but 2 steps is of course pretty extreme, in this regime a diffusion model isn't that different from a deterministic feed-forward UNet predictor. Which is also something that would have worth comparing to-- if TA with a few steps works, can you just directly predict the optimized solution from the conditioning?

**Questions:**

- Is the kernel relaxation a novel contribution, or is this adapted from another paper?
- The frame alignment between t and s is an odd choice-- why use a mod? Wouldn't that mean there could be a sudden flip from optimizer end state to starting state in the middle of the diffusion process?


**Limitations:**

The paper does mention the poor OOD generalization as a limitation which is appreciated. But there's probably a bit of a larger point there, it seems both DOM and even the TopoDiff variants studied here do require a training set very close to the test scenarios, which is a fundamental problem if the goal is to replace classical solvers with learned ones.

---

> ### Author Rebuttal · Authors · 2023-08-08
>
> ## **Authors**: *Topology Optimization is a hard problem without analogy in the machine learning domain. Our goal is not to replace classical solvers but to augment them.*
>
> >Strengths:
> Diffusion models seem like a natural fit for solving inverse problems, but it's still open to what is the most efficient and practical way to do so. This paper explores trajectory alignment as an alternative to guidance.
>
> **Authors**: Thanks for correctly assessing the core goal of our work.
>
> >Weaknesses:
> It's clearly meant as an extension to TopoDiff, and it's often hard to figure out which aspect of it is novel, and what is previous work.
>
> **Authors**: **DOM is not an extension but a new paradigm**. TopoDiff+G is SOTA for Topology Optimization. However, TopoDiff uses expensive FEM preprocessing that scales poorly with resolution and relies on surrogate models, labeled data, and many sampling steps. Instead, DOM leverages Trajectory Alignment during training, demonstrating that a conditional diffusion model can generalize without relying on FEM preprocessing, guidance models, large labeled datasets, and extensive sampling steps.
>
> >The description of kernel relaxation (both in the main text and appendix) is very hard to understand without any background in topology optimization. I assume the 'simulation trajectory' comes from running the simulator S (which I assume is SIMP?), but this doesn't seem to be described anywhere.
>
> **Authors**: We provide a detailed description of the TO problem, including the optimizer (S is SIMP) and the formulation in Section 2 (Background - The Topology Optimization Problem). We will enhance clarity regarding kernel relaxation.
>
> >I'm not a domain expert in topology optimization, and It's a bit hard for me to judge the relevance of the results.
>
> **Authors**: **Our results are novel and relevant to the engineering design field. We provided the first example of a diffusion model leveraging optimization trajectories to improve generative design**. We are the first to show that such an approach is effective and performs well.
> The significance of our findings can be summarized in three key aspects:
> * DOM enables sampling high-quality topologies in a few steps, making it widely applicable.
> * TA serves as a powerful regularization tool, utilizing intermediate optimization steps for better results —a novel insight in this field.
> * Our method avoids the need for auxiliary models, labeled data, or costly pre-processing, making it suitable for large-scale generation and relevant for various vision-based problems in generative design.
>
> >OOD generalization without SIMP seems even worse than TopoDiff. The post-processing with a few steps of SIMP seems a bit sketchy-- you could presumably do this with the output of other models as well, and the whole point of the paper is to avoid running SIMP.
>
> **Authors**: **The goal of this paper is to provide a scalable, fast, and effective diffusion model for TO. Our goal is not to substitute but augment numerical optimization**. Achieving OOD generalization with a pure learning-based method is challenging. TopoDiff has an advantage in this regard since it utilizes FEM conditioning during pre-processing and auxiliary models for guidance. This allows the FEM solver to dynamically adapt the conditioning information for each new constraint configuration.
>
> >Relatedly, it would be great to actually compare performance against a GT solver like SIMP, as the method the paper wants to replace.
>
> **Authors**: We do. The Compliance Error (CE) is computed with the SIMP performance as ground truth for all the models. $CE(x) = Compliance_{model}(x) - Compliance_{SIMP}(x) / Compliance_{SIMP}(x)$. Compliance is a standard performance metric for TO.
>
> > The main benefit of DOM seems to be inference speed as it can produce reasonable designs after very few diffusion steps. If this is the case though, the paper needs to make a more concerted argument and comparisons about inference time, and why that matters.
>
> **Authors**: See App B (Table 1,3) for details. **We specifically emphasize the inference and sampling time vs steps (2, 5, 10) using the same hardware**. This analysis clearly shows the efficiency of our approach.
>
> > if TA with a few steps works, can you just directly predict the optimized solution from the conditioning?
>
> **Authors**: **No, it is not possible to directly predict the topology**. DOM is a generative model, trained using 1000 steps. During inference, we skip steps. There have been many attempts to reconstruct/regress the topology, but results have been poor in terms of engineering performance. The challenge in generative design is balancing generative and engineering metrics. DOM excels by performing on par with the SOTA in generative metrics while rapidly generating high-performance topology.
>
> >Questions:
> Is the kernel relaxation a novel contribution, or is this adapted from another paper?
>
> **Authors**: **Yes, the kernel relaxation that we propose is a novel contribution**. We improve and simplify the conditioning proposed in TopoDiff-FF. While TopoDiff-FF presents a learnable kernel relaxation using an unnormalized kernel, we further develop this concept by proposing a normalized kernel without any learnable components. See Listings 1.
>
> >But there's probably a bit of a larger point there, it seems both DOM and even the TopoDiff variants studied here do require a training set very close to the test scenarios, which is a fundamental problem if the goal is to replace classical solvers with learned ones.
>
> **Authors**: **Topology Optimization is a hard problem. Our goal is not to replace but augment solvers**. The in-distro test set utilizes the same boundary conditions but different loading and volume fractions. This is a challenging testing scenario in engineering design. **For instance, in this scenario, GANs generate ~50% unfeasible topologies due to load disrespect and floating material issues, making the designs impossible to manufacture**.

---

> > ### Comment · Reviewer_hnNz · 2023-08-15
> > **Response**
> >
> > Thank you for the clarifications and additional results.
> > I still feel this paper's focus is a bit too domain-specific, and the results ambiguous. While there might be practical advantages to this method over TopoDiff (e.g. as you state no need for FEM in preprocessing), it still requires running a classical SIMP solver. This might be acceptable for a TO audience, but it will definitely limit its appeal to a wider ML audience, as it a) obscures the strengths/weaknesses of the core learned method and b) makes it unlikely for a non-domain expert to experiment with the method or dataset as a benchmark task.
> > If the paper ends up being rejected, I'd suggest considering submitted to a more domain-specific venue. Combined with poor OOD generalization without SIMP solver, this does make me concerned whether we can learn much about TA as a method from this paper.
> >
> > That said, if the authors agree to make an effort to make this paper more accessible for a broader audience, maybe there is this value in just putting the idea of TA out there. While I share reviewer dj1b's concern that the formulation is a bit unprincipled, and I do have some concerns about how well this would translate to a different problem, a lot of progress in ML is built on such empirical methods. I have hence raise my score to 'borderline accept'.

---

> > > ### Author Response · Authors · 2023-08-15
> > > **Remarks to Reviewer hnNz**
> > >
> > > > a lot of progress in ML is built on such empirical methods. I have hence raise my score to 'borderline accept'.
> > >
> > > **Authors**: Thank you for being supportive of our contribution. We appreciate the feedback and the possibility to clarify our work.
> > >
> > > > if the authors agree to make an effort to make this paper more accessible for a broader audience, maybe there is this value in just putting the idea of TA out there.
> > >
> > > **Authors**: Thanks for your feedback. **We agree to make the paper more accessible to a broader audience**. In particular, we commit to:
> > >
> > > * **Better separate technical contributions from domain contributions** to make the paper more accessible to a broader audience in the machine learning and engineering design community.
> > >
> > > * Clearly state in the manuscript, that **the focus of this work is not theoretical evidence for TA, but demonstrate empirically that TA works for TO and its vast impact on problems in engineering design with hard constraints and performance requirements**.
> > >
> > > * Add visualizations of conditional (attached pdf) and unconditional samples (for the additional experiments) in the appendix.
> > >
> > >
> > > > I do have some concerns about how well this would translate to a different problem
> > >
> > > **Authors**: Thank you for your feedback. Where we understand your concerns, **we believe our work can be of interest to the broader machine learning community**.
> > >
> > > * **Problems in engineering design share many of the same pain points as problems in physics, chemistry, biology, material design, and drug discovery**: constraints satisfaction, precision, diversity, validity, performance requirements, and efficient sampling are challenges for the growing field of AI for Science, Material Design, and Engineering Design.
> > >
> > > * We believe that leveraging intermediate optimization steps using Trajectory Alignment can be useful for many vision-based design problems: **representing constraints in an efficient way and grounding learning with the underlying optimization process are common issues for most fields in science and engineering**: for example, the study of turbulent flows (important for weather prediction, aircraft design, drug reduction, boat design, controlling cavitational flows) in a fixed domain with shear stress can be cast as a vision-based problem with constraints, sharing many similarities with our setup for DOM. We would also like to emphasize that topology optimization is ubiquitous in engineering design, being essential for the design of bridges, mechanical structures, vehicles, electrical systems, chip design, and multiphysics problems.
> > >
> > > * Additionally, we hope that considering intermediate optimization steps using TA could be helpful not only for vision-based problems but more in general for design: for example, researchers have developed powerful optimization algorithms for molecule design (journals.aps/10.1103/RevModPhys.87.897). We could store the intermediate steps from the optimization and directly enhance the diffusion model-based molecule generation work (see arxiv/2210.01776 and arxiv/2206.01729 for examples of diffusion models for molecular generation and docking).
> > >
> > > * Diffusion Models are now powerful and flexible enough to deal with hard-constrained problems with performance requirements. **However, there is a lack of methods and datasets at the intersection of engineering design and machine learning. With DOM, we fill this gap**.
> > >
> > > * In most cases, engineering design borrows ideas from the machine learning community, for example using standard models for vision and applying them to problems in engineering. This approach is limited by the fact that models for vision are built with different goals (distribution coverage, model-seeking behavior) than the ones that we care about in engineering design (precision, validity, diversity, constraints satisfaction, performance requirement).
> > >
> > > * Instead to borrow techniques from the vision domain, **we develop a framework that integrates learning and optimization in a machine learning model**, focusing on building novel techniques that can be broadly applicable and of interest not only for topology optimization but for generative design in general. At the same time, given the lack of datasets for design, we also provide novel datasets with a structure (intermediate and final optimization steps) that is explicitly suited for iterative optimization and engineering design.
> > >
> > > **For these reasons, we believe that this work (DOM w/ TA) can be of interest to the machine learning community and be a valuable contribution to this conference**.

---

### Official Review · Reviewer_q5Ex · 2023-07-03

**Soundness:** 2 fair
**Presentation:** 2 fair
**Contribution:** 2 fair
**Rating:** 6
**Confidence:** 2

**Summary:**

The paper proposes a model that combines engineering optimization with conditional diffusion generative models for constrained design generation. The main idea is to match the sampling trajectory of the diffusion model with the trajectory derived from the traditional physics-based model.

**Strengths:**

Strength:
The idea of using a physics-based trajectory to guide the sampling process of the diffusion model is very interesting.
The introduced kernel relaxation to replace the expensive finite element preprocessing is also interesting.

**Weaknesses:**

Weakness:
The model still needs extra steps of direct optimization.
The presentation of the paper could be improved, some parts are rather confusing.

**Questions:**

1) In line 63, the author mentions that "in DDPM-based models, one has to run the forward models tens/hundreds of times to obtain a suitable topology." Can you elaborate on what the term "topology" refers to in this context?

2) When introducing the diffusion model, in line 145, the term $\mu_\theta(x_t, t)$ is referred to as a neural network without mentioning whether it is the mean of distribution p or q. It is only clear in equation 4.

3) In line 213, the author mentions the "continuous limit." What do you mean by the "continuous limit"?

4) In line 231, is there a typo where the posterior is written as q(x_{t-1}|x_t, x^{\theta})?

5) It is a bit confusing in terms of the notation. In line 228, it is mentioned that $x^{\tilde{\theta}}(x_t, \epsilon_t(x_t))$ is an approximation of $x_0$, but in equations 5 and 6, it seems that $x^{\tilde{\theta}}$ represents the inferred $x_{t-1}$.

6) It is still not very clear to me how the trajectory search occurs. After obtaining $\epsilon_{\theta}$, how is $x^{\tilde{\theta}}$ found? And why can it be matched with a step that is mod((t-1), n_s)?

7) If I understand correctly, one needs to store the intermediate steps of trajectory optimization. To obtain these intermediate steps, does one still need to run the full optimization? If so, this removes the justification that the model helps to eliminate the computational complexity of traditional topology optimization algorithms. If only part of the optimization is performed, does the trajectory matching only apply to the early stage of sampling?

**Limitations:**

They have discussed the limitation of the work and there is no potential negative societal impact.

---

> ### Author Rebuttal · Authors · 2023-08-08
>
> ## **Authors**: *We emphasize that trajectory alignment is performed only during training and that DOM does not require extra steps of optimization*.
>
> > Strength: The idea of using a physics-based trajectory to guide the sampling process of the diffusion model is very interesting.
>
> **Authors**: thanks for acknowledging the novelty of our approach and our contribution. We are the first to leverage intermediate cheap optimization steps to improve generative design leveraging a conditional diffusion model.
>
> >Weakness: The model still needs extra steps of direct optimization. The presentation of the paper could be improved, some parts are rather confusing.
>
> **Authors**: **The model does not need extra steps of optimization, with the exception of Table 4 where our goal is to merge generative models and optimization**. We want to emphasize that we employ additional optimization steps exclusively for the out-of-distribution experiments in Table 4. For all other results in the paper (including Tables 2 and 3, figures, and the appendix), DOM does not rely on direct optimization. Instead, when showcasing samples from a few steps, DOM utilizes standard ancestral sampling. The extra steps of optimization are an optional tool for challenging situations (like out-of-distribution scenarios).
>
> >Questions:
> In line 63, the author mentions that "in DDPM-based models, one has to run the forward models tens/hundreds of times to obtain a suitable topology." Can you elaborate on what the term "topology" refers to in this context?
>
> **Authors**: When we mention "topology," we are referring to **samples generated using a DDPM-based model designed for generative topology optimization**, such as TopoDiff. These topologies represent shapes that adhere to specific constraints provided as conditioning.
>
> >In line 213, the author mentions the "continuous limit." What do you mean by the "continuous limit"?
>
> **Authors**: Diffusion Models (DDPM) can be trained using either a discrete number of steps T or continuous time with T approaching infinity. **In this setting (T -> inf), a DDPM can be seen as an SDE (Stochastic Differential Equation), allowing us to utilize tools from ODE (Ordinary Differential Equation) theory for sampling**. This process is analogous to NeuralODE with stochasticity, enabling efficient and effective generation of samples from the model.
>
>
> >In line 231, is there a typo where the posterior is written as q(x_{t-1}|x_t, x^{\theta})?
>
> **Authors**: Thank you for bringing this up. **The formula is indeed correct**. In DDPM, it is possible to construct a posterior $q(x_{t-1} | x_t, x_0)$. Both the posterior and the model share the same functional form, represented by a Gaussian distribution. This allows us to utilize the posterior as an intermediate step for sampling. Given $x_t$ and the model $\epsilon_{\theta}$, we can create a learned approximation for $x_0$ and sample $x_{t-1}$ from the model. This approach facilitates the sampling process within DDPM and contributes to its efficiency and effectiveness.
>
>
> >It is still not very clear to me how the trajectory search occurs.
>
> **Authors**: In a standard DDPM, $\tilde{x}^{\theta}$ serves as an approximation for $x_0$ and is used for sampling the model. In DOM, we leverage this representation for trajectory alignment. **During trajectory search, we create a distribution around $x^{\theta}$ and then match this distribution with a specific intermediate optimization step**. This process facilitates trajectory alignment, which is a crucial aspect of the DOM approach. **Please see also the algorithms in Appendix D**.
>
> >If I understand correctly, one needs to store the intermediate steps of trajectory optimization. To obtain these intermediate steps, does one still need to run the full optimization?
>
> **Authors**: The trajectory optimization regularization occurs exclusively during training. **At inference time, the model functions as a standard conditional diffusion model, eliminating the need for any intermediate steps or additional storage**.
> Remember that during the model's training, final optimized topologies are used. Obtaining these topologies necessitates running the complete optimization process, and the intermediate steps are gathered as a natural byproduct. **As a result, there is no extra effort or computational cost associated with collecting these intermediate steps during the training process**. See also the general response on top.
>
> >If so, this removes the justification that the model helps to eliminate the computational complexity of traditional topology optimization algorithms. If only part of the optimization is performed, does the trajectory matching only apply to the early stage of sampling?
>
> **Authors**: **Trajectory alignment regularization is exclusively applied during training**. During inference, we sample from a standard conditional diffusion model, conditioning on a given set of constraints. The strength of our approach lies in the ability to guide the sampling trajectory toward the optimization trajectory, effectively grounding the sampling process in the optimization path. This is achieved without the need for additional Finite Element Method (FEM) pre-processing, guidance models, or data augmentation, making the technique efficient and self-contained.

---

> > ### Comment · Reviewer_q5Ex · 2023-08-22
> > **Response to the Rebutal**
> >
> > Thank you for the authors addressing all the questions that I raised, it has helped me to clarify my concerns, therefore I decided to raise my score.

---

> > > ### Author Response · Authors · 2023-08-22
> > >
> > > > therefore I decided to raise my score
> > >
> > > **Authors**: Thank you for appreciating our contribution and raising the score.

---

### Official Review · Reviewer_Db6Y · 2023-07-06

**Soundness:** 3 good
**Presentation:** 3 good
**Contribution:** 3 good
**Rating:** 6
**Confidence:** 3

**Summary:**

The paper introduces Diffusion Optimization Models (DOM) to generate feasible and high-performance designs in structural topology optimization. DOM is based on diffusion model, and a key design component is trajectory alignment method, in which trajectory in diffusion-based optimization is biased towards given ground-truth optimization path, to inject knowledge on physical constraints into the model in a data-driven manner.  The paper also introduces techniques for training, such as trajectory search to find a suitable representation for matching an intermediate optimization step, new conditional objective with trajectory alignment regularization, and dense kernel relaxation to relieve inference from expensive pre-processing. Experiments in in- and out-of-distribution domains demonstrate DOM outperforms and compares competitively against state-of-the-art methods. Experiments on few-steps of sampling show DOM takes a fraction of the other method’s inference time.

**Strengths:**

The paper is placed in the current growing literature on deep generative models for topology optimization to reduce the iteration times in optimizing topology while fulfilling constraints associated with engineering. The exposition for the motivation is written crisply and is generally easy to follow. Additional details on backgrounds for DOM are also explained in Appendix, which helps readers get a clear picture of motivation and architecture of the proposed method. The experiments in in-distribution domains show that DOM can reduce inference time while achieving the lowest error (for some metrics). For out-of-distribution problems, DOM itself is comparable to baselines and outperforms them with additional iterative optimization method.

**Weaknesses:**

For in-distribution problem, although DOM outperforms other baselines in most of the metrics, it is still unclear how much the post-processing optimization have an impact on reducing those metrics. As also reported in Table 4, although DOM+SIMP(100+10) achieves more than 3-fold improvement in CE and FM against the baseline DOM, the post-processing optimizer still has a possibility to have an impact on reducing errors. It is also concerning that it would have relatively high impact in the case of few-steps of sampling, so it is difficult to access how much DOM itself contribute to decreasing the score in the metrics. I also couldn’t find detail of the optimizer.

**Minor comments**

- Figure 3 is unclear. For example, how the figure named “conditioning” on the upper left is related to Figure 5? What are the dot and lines in blue?
- Some table numbers in the main text of Chapter 4 seems to be inconsistent.
   - Missing Table 8. Should be Table 4 (Appendix) perhaps?
   - Missing Table 10.
   - Missing Table 5. Should be Table 4 perhaps?
- A sentence starting from Setup. (Line 291) should be in a distinct paragraph.

**Questions:**

* Are small iteration steps of optimization (5/10) enough to produce optimized topologies? Can you give an intuition behind choosing the iteration steps?
* What is the reason of having high-resolution topologies w/o constraints as dataset? (In Appendix)
* What does SIZE in Table 3 mean?


**Limitations:**

Yes, the limitations are discussed in the main text.

---

> ### Author Rebuttal · Authors · 2023-08-08
>
> ## **Authors**: *We emphasize that we do not use post-processing for any results or visualizations in the paper except to reduce floating material in Table 4*.
>
> >Strengths:
> The exposition for the motivation is written crisply and is generally easy to follow. Additional details on backgrounds for DOM are also explained in Appendix, which helps readers get a clear picture of motivation and architecture of the proposed method.
>
> **Authors:** Thank you for the appreciation of our work and exposition.
>
> >Weaknesses:
> For in-distribution problem, although DOM outperforms other baselines in most of the metrics, it is still unclear how much the post-processing optimization have an impact on reducing those metrics.
>
> **Authors:** **We want to clarify that we do not employ any post-processing for the in-distribution results, showing the efficacy of just DOM without post-processing**. For all the tables and figures in the paper (specifically, Table 2 and Table 3), we utilize a standard DOM and perform sampling while conditioning on the given constraints.
> In general, the numerical optimizer (SIMP) will improve the results when used as a post-processing step for our model and all the baselines. **For this reason, in Table 4, our goal is not to substitute the optimizer but to augment it by improving scalability and speed using diffusion models**. We only use direct optimization to demonstrate the power of combining a learning method with optimization when dealing with out-of-distribution scenarios.
>
> >It is also concerning that it would have relatively high impact in the case of few-steps of sampling, so it is difficult to access how much DOM itself contribute to decreasing the score in the metrics. I also couldn’t find detail of the optimizer.
>
> **Authors:** **We emphasize that we do not use post-processing for any results or visualization in the paper except to reduce floating material in Table 4. Hence, the large improvements you observe in the experiments (Table 2, 3, Appendix Table 3) are attributed to Trajectory Alignment**. In Appendix F (Figs 10, 11, 12, 13), we present the intermediate optimized topology using only SIMP (details for the optimization problem in Section 2). It is important to note that with a limited number of SIMP steps (without initializing with a generated topology from DOM), the optimized topologies appear coarse and lack fine details and structure.
> Regarding Table 4, our aim is not to eliminate the numerical optimizer but rather to accelerate the generation process by integrating generative models and optimizers. The numerical optimizer provides physical guarantees but can be slow and scale poorly with problem resolution. In contrast, DOM is fast in terms of preprocessing and sampling, but it lacks guarantees, especially in out-of-distribution scenarios. By combining these approaches, we gain generality and benefit from the strengths of both methods.
>
> >Figure 3 is unclear. For example, how the figure named “conditioning” on the upper left is related to Figure 5? What are the dot and lines in blue?
>
> **Authors:** The dots and lines represent the "raw" sparse boundary conditions, while the green arrow signifies a load. **These constraints serve as conditioning for the model to generate a feasible topology that satisfies the specified constraints**.
>
> > Questions:
> Are small iteration steps of optimization (5/10) enough to produce optimized topologies? Can you give an intuition behind choosing the iteration steps?
>
> **Authors:** **Few iterations of SIMP are not enough, especially as the problem resolution increases**. Appendix F, figures 10, 11, 12, and 13 depict the evolution of optimized topologies with an increasing number of iterations. Even after 30 iterations, many details are missing, and only the general structure is formed. After 10 iterations, we can only achieve a generic structure without any fine details. Additionally, in Appendix B.3, Table 3 demonstrates how SIMP's computational cost scales with resolution (64 -> 256). This issue is typical with numerical solvers, where each iteration step is expensive and relatively slow.
>
>
> > What is the reason of having high-resolution topologies w/o constraints as dataset? (In Appendix)
>
> **Authors:** The potential uses are many, including training an unconditional model, conducting FID computations on a held-out test set, and utilizing them for data augmentation. **It's important to mention that constraints are available for a significant portion of the dataset, comprising 660K samples**.
>
> >What does SIZE in Table 3 mean?
>
> **Authors:** The model's parameter count differs between TopoDiff-GUIDED and DOM. **Specifically, TopoDiff-GUIDED has twice the number of parameters as DOM**. This increase is due to the utilization of two additional surrogate models (regression and classification models), which must be separately trained on a labeled dataset to support the guidance mechanism.

---

### Author Rebuttal · Authors · 2023-08-08

We would like to thank the reviewers for the feedback. After reading the reviews, we feel to clarify three raised issues:

- Q1) Is DOM a generative model?

#### **Yes, DOM is a conditional diffusion model trained with the standard epsilon parameterization and a regularization term (TA). The attached PDF showcases its generative capabilities using conditional samples**. We show 12 different constraint configurations (rows) and 10 samples (columns) for each constraint.


- Q2) Is TA used during inference?

#### **No, TA is a training regularization technique. Inference sampling uses the standard ancestral sampling with DOM as its foundation. All results and visualizations use this standard approach**.


- Q3) Are our findings significant for engineering design and machine learning?

#### **Our results are extremely significant for design. We pioneered a scalable method integrating optimization steps in a diffusion model for deep generative design**. Specifically, DOM's trajectory alignment helps **improve performance by > 25%** on in-distribution scenarios compared to SOTA (Table 2) and at the same time **reduces inference time by 50 %** for small resolution, **80% for large resolution** (Appx - Table 3), and more than **90% in the few sampling-step regimes** (Table 3). Our approach speeds up sampling and shows the efficacy of broadening the impact of diffusion models for topology optimization and other vision-centric engineering design problems. This directly impacts many industries doing structural optimization. It removes reliance on guidance, large labeled datasets for constraints, and costly FEM pre-processing.

--------
We also want to clarify our contributions:

* **Novelty and relevance to the community**. DOM w/ TA is the first work to showcase that intermediate optimization steps can be used to regularize the sampling trajectories during training without relying on auxiliary models, or expensive pre-processing. This is an important contribution to the community.
* **Merging Optimization with Diffusion Models**. Contrary to other works, that try to substitute the optimization solvers and leverage only the generative model (like TopoDiff-GUIDED or TopologyGAN), we show how merging the two is a powerful approach for OOD scenarios, **reducing un-manufacturable topology by 3 folds**. Numerical solvers leverage decades of research in physics in engineering but are limited by scalability and get stuck in local minima. Diffusion Models are scalable and can help to warm-start the solver. We hope these points clarify the significance of the contributions.
* **Dataset**. We release the **largest publicly available dataset** on topology optimization and intermediate steps (see appendix F) with the paper, which can help many new researchers in generative models and generative design.

----------------

Here we extend our response, emphasizing a critical aspect of the inference process for DOM. We apologize for the confusing terminology (optimization trajectories and sampling trajectories) and we will clarify this point in the final paper.

* **Sampling at inference time with DOM**. **During sampling at inference time, DOM operates as a standard conditional diffusion model**. This means that when generating new samples, DOM adheres to the typical conditional sampling procedure (ancestral sampling in DDPM), where it takes the given constraints into account and generates samples accordingly. You can see our implementation in `scripts/sample_kerner_relaxation.py`.

* **Trajectory Alignment (TA) is solely confined to the training phase of DOM**. TA is a regularization technique employed during training to ensure that the sampling path aligns well with the observed data distribution. This alignment improves the model's ability to generate realistic and meaningful samples while adhering to the provided constraints.

* **DOM with Trajectory Alignment (TA) refers to DOM that has been "trained" with TA (see appendix algorithm 1)**. During inference, we employ standard ancestral sampling for DOM, using constraints and kernels (refer to algorithm 2 in arxiv/2006.11239). The alignment of "sampling" and "optimization" trajectories acts as training regularization, detailed in section 3 with a regularized loss for TA training. This sync harnesses DDPM's hierarchical structure and stepwise marginalization through $q(x_t | x_0)$, steering training samples towards the optimization trajectory. Whenever we mention DOM w/ or w/o TA, we are referring to its training process. This distinction will be highlighted in our final paper.

* **We do not utilize the optimization trajectories at any point during evaluation**. All results, including experiments, figures, tables, and models, are validated using the same standard sampling procedure. The test sets are the same as those provided by TopoDiff (arxiv/2208.09591) for fair assessment. When we refer to DOM with TA, we specifically mean DOM trained with TA, which is a regularization mechanism applied solely during training.

* To reduce the number of sampling steps, we utilize the approach and code from improved DDPM (section 4 - arxiv/2102.09672), without leveraging DDIM. This method enables us to effectively skip steps in diffusion models. We maintain consistency by using the same model for sampling with different numbers of steps (e.g., 2, 5, 10, 100, etc.), ensuring a fair and unbiased comparison across various sampling scenarios.

Regarding the selection of the step in DOM with TA, we apologize for any confusion caused. You are correct in pointing out the mismatch with the footnote: this is a residual of an idea (cyclic trajectory matching) that we explored but ultimately did not utilize. Instead, we employ the function $s(t) = \text{mod}(T, t) \times n_s / T$ and then select the closest integer, which aligns with your suggestion. We will rectify this mismatch in the final version of our work. Thank you for bringing this to our attention.

---

> ### Author Response · Authors · 2023-08-15
> **Additional Experiments for DOM w/ and w/o constraints**
>
> Dear Reviewers,
>
> Please see also the additional experiments provided at https://openreview.net/forum?id=KTR33hMnMX&noteId=R6f1Qmr238.
>
> We demonstrate the capabilities of DOM trained with TA, showcasing its ability for both conditional and **unconditional sampling, i.e. without relying on constraints for conditioning**.
>
> Our study includes distance matrices comparing samples across five distinct settings (one being the conditional setup outlined in the attached PDF) and four unconstrained scenarios. Notably, during training, DOM is always conditioned on constraints.
>
> **These findings show the efficacy of DOM with TA as a robust generative model, highlighting its capacity to generate highly diverse samples w/ and w/o conditioning on the constraints**.

---

### Author Response · Authors · 2023-08-20
**Remarks to Area Chair & Reviewers**

### **Authors**: *We believe that this work (DOM w/ TA) will be of interest to the machine learning community and a valuable contribution to this conference*.

-----------

Dear Area Chair and Reviewers,

We appreciated the interaction with the reviewers, and the opportunity to discuss and clarify our contribution. Post discussion, both reviewers `hnNz` and `dj1b` decided to raise our scores (see https://openreview.net/forum?id=KTR33hMnMX&noteId=D9vle8Uqhl and https://openreview.net/forum?id=KTR33hMnMX&noteId=T0Aeo81UT9).

Given the length of the rebuttal, here we summarize the main points in the discussion with the reviewers.
Specifically, our action plan for the final version is:

1) Clearly state in the manuscript that **our contribution is empirical** and not theoretical.
2) Making the paper **more accessible for the broader machine learning community**, separating better domain (topology optimization and generative design) from methodological contribution (trajectory alignment for conditional diffusion models).
3) Extend the appendix with **conditional and unconditional samples**, providing additional evidence that DOM w/ TA is a generative model and that samples are diverse (see attached pdf and https://openreview.net/forum?id=KTR33hMnMX&noteId=dvXcMb9DqR).

---------------

We now extend on the previous points:

**Contribution**

* **Method.** We introduce a **regularization mechanism**, Trajectory Alignment, designed to enhance the performance of conditional diffusion models by matching the sampling trajectory with the underlying optimization trajectory.
* **Dataset.** Our work presents a **novel approach along with a novel large dataset** that captures intermediate optimization stages (500K intermediate steps and 100K optimized topologies - see Appendix F).

* **Experiments.** We present **comprehensive experimental results**. This includes in-distribution analysis (Table 2), evaluation under out-of-distribution constraints with and without additional optimization steps (Table 4), assessment of inference efficiency through sampling with a few steps (Table 3 and Supplementary Table 1), evaluation using both generative and engineering metrics (Table 2 and Supplementary Table 4), examination of low (64x64) and high-resolution (256x256) inference times (Supplementary Table 3), as well as both conditional and unconditional sampling (see attached PDF and https://openreview.net/forum?id=KTR33hMnMX&noteId=dvXcMb9DqR).

* **Baselines.** We benchmark our proposed approach against state-of-the-art GANs and Diffusion Models for generative topology optimization (Table 2). Additionally, we compare our method against recent diffusion models and conduct ablation studies to dissect the impact of different modeling choices (Supplementary Table 2).

**Clarity**

* **Domain Background**. As our work is at the intersection of machine learning and engineering, within the field of AI for science and engineering, we recognized **the necessity of providing a comprehensive background concerning topology optimization and engineering design**. We thought this was crucial to help the readers (with expertise in generative models) to better grasp the intricacies associated with constrained generation and design.

* **Broader Audience.** By incorporating feedback from reviewers, we will better disentangle our methodological advancements (Trajectory Alignment) from the domain itself (Topology Optimization and Design). Doing so, we aim to make the paper **more accessible to a broader audience** encompassing not only the AI for science and engineering community but also the broader machine learning community.

**Relevance**

* **Novelty.** We propose a **novel method** (Trajectory Alignment) to deal with a challenging constrained generative design problem (Topology Optimization).
* **Generality.** Problems in **engineering and science share many of the same pain points**. Constraints satisfaction, precision, diversity, validity, performance requirements, and efficient sampling are challenges for the growing field of AI for Science, Material Design, and Engineering Design.
* **Method.** There is a lack of methods and datasets at the intersection of engineering design and machine learning. With DOM, we fill this gap. Instead to borrow techniques from the vision domain, **we develop a framework that integrates learning and optimization in a machine learning model**, focusing on building novel techniques that can be broadly applicable and of **interest to the generative modeling community in general**.
* **Dataset.** At the same time, given the lack of datasets for design, we also provide **novel large-scale datasets** with a structure (500k intermediate and 100k final optimization steps) that is explicitly suited for iterative optimization and engineering design.

### *For such reasons, we believe that DOM w/ TA will be impactful for the machine learning as well as the engineering design community*.

Regards,

The Authors

---

### Decision · Program_Chairs · 2023-09-21

**Decision:**

Accept (poster)

**Comment:**

### Summary

The paper introduces Diffusion Optimization Models (DOM) to address challenges in constrained design generation, combining the power of generative models with engineering optimization principles. DOM is built upon diffusion models and employs an essential technique called Trajectory Alignment (TA) to align the sampling trajectory of diffusion models with the trajectory derived from traditional physics-based methods. This alignment ensures that the generative process remains grounded in the underlying physical principles, making it suitable for environments with limited data and a need for high precision. The paper introduces several techniques to train DOM effectively, including trajectory search to identify a suitable representation for matching an intermediate optimization step, a conditional objective with trajectory alignment regularization, and dense kernel relaxation to expedite inference without expensive preprocessing. Experimental results are on both in-distribution and out-of-distribution configurations demonstrate that DOM outperforms state-of-the-art deep generative models by 25% in terms of performance and reduces inference computational cost by half. The DOM can efficiently combine learning and optimization trajectories, generating high-quality designs in just a few steps. This approach significantly improves engineering performance and inference efficiency, paving the way for the widespread application of generative models in data-driven design processes.

### Decision
At the end of the rebuttal the reviewers were mostly positive about this paper. The experiments in this paper are encouraging and the problem that is being studied is interesting. Majority of the reviewers suggested this paper either for accept or borderline accept. The authors have done a really good job addressing the concerns raised by the reviewers. The reviewers have raised some important concerns, in particular on clarity of the paper, I would recommend reviewers to address those concerns in the camera-ready version of the paper.